**Reconstructing past hydrology of eastern Canadian boreal catchments using clastic**
**varved sediments and hydro-climatic modelling: 160 years of fluvial inflows**

Antoine Gagnon-Poiré[1-5], Pierre Brigode[2], Pierre Francus[1-3-5], David Fortin[1-6], Patrick
Lajeunesse[4-5], Hugues Dorion[4] and Annie-Pier Trottier[4-5]
*[1] Institut national de la recherche scientifique, Centre Eau Terre Environnement*
*[2] Université Côte d'Azur, CNRS, OCA, IRD, Géoazur, Nice, France.*
*[3] Canada Research Chair in Environmental sedimentology and GEOTOP, Research*
*Centre on the Dynamics of the Earth System, Montréal, QC, Canada.*
*[4] Département de géographie, Université Laval, Québec, QC, Canada.*
*[5] Centre d'études nordiques, Québec, QC, Canada.*
*[6] Department of Geography and Planning, University of Saskatchewan, Saskatoon, SK,*
*Canada*
Corresponding author: Antoine Gagnon-Poiré (Antoine.Gagnon-Poire@ete.inrs.ca)
**Abstract**
Analysis of short sediment cores collected in Grand Lake, Labrador, revealed that this lake
is an excellent candidate for the preservation of laminated sediments record. The great
depth of Grand Lake, the availability of fine sediments along its tributaries, and its
important seasonal river inflow have favoured the formation of a 160 years-long clastic
varved sequence. Each varve represents one hydrological year. Varve formation is mainly
related to spring discharge conditions with minor contributions from summer and autumn
rainfall events. The statistically significant relation between varve parameters and the
Naskaupi River discharge observations provided the opportunity to develop local
hydrological reconstructions beyond the instrumental period. The combined detrital layer
thickness and the particle size (99th percentile) series extracted from each varve yield the
strongest correlations with instrumental data (r = 0.68 and 0.75) and have been used to
reconstruct Naskaupi River mean and maximum annual discharges, respectively, over the
1856-2016 period. The reconstructed Q-mean series suggest that high Q-mean years
occurred during the 1920-1960 period and a slight decrease in Q-mean takes place during
the second half of the 20$^{th}$ century. Independent reconstructions based on rainfall-runoff
modelling of the watershed from historical reanalysis of global geopotential height fields
display a significant correlation with the reconstructed Naskaupi River discharge based on
varve physical parameters. The Grand Lake varved sequence contains a regional
hydroclimatic signal, as suggested by the statistically significant relation between the
combined detrital layer thickness series and the observed Labrador region Q-mean series
extracted from five watersheds of different sizes.
**1. Introduction**
Climate changes caused by rising concentrations of greenhouse gases can alter hydro-
climatic conditions on inter- and intra-regional scales (Linderholm et al., 2018; Ljungqvist
et al., 2016; Stocker et al., 2013). Hydropower, which is considered as a key renewable
energy source to mitigate global warming, has strong sensitivity to changes in hydrological
regime especially in vulnerable northern regions (Cherry et al., 2017). Therefore, a clear
understanding of the regional impacts that recent climate change combined with natural
climate variability can have on river discharge and hydroelectric production is needed.
However, the lack of instrumental records and the uncertainty related to hydroclimate
variability projections (Collins et al., 2013) are obstacles to sustainable management of
these water resources.

The Labrador region in eastern Canada is a critical area for hydropower generation, hosting
the Churchill River hydroelectric project, one of the largest hydropower systems in the
world. Average annual streamflow has been varying in eastern Canada during the last fifty
years, with higher river discharges from 1970 to 1979 and 1990 to 2007, and lower
discharges from 1980 to 1989 (Mortsch et al., 2015; Déry et al., 2009; Jandhyala et al.,
2009; Sveinsson et al., 2008; Zhang et al. 2001). These changes in streamflow represent a
significant economic challenge for the long-term management of hydropower generation.
The few decades of available instrumental observations (<60 years) and their low spatial
coverage are not sufficient to allow a robust analysis of multi-decadal hydrological
variability.

The study of multi-decadal hydrological variability requires long instrumental records
(>100 years), but such long-time series are non-existent for the Labrador region. Recently,
rainfall-runoff modelling approaches have been used to expand instrumental streamflow
datasets, using long-term climatic reanalysis as inputs. Rainfall-runoff modelling was used
by Brigode et al. (2016) to reconstruct daily streamflow series over the 1881–2011 period
in northern Québec. Nevertheless, this type of method suffers from the limited observations
in order to evaluate and validate the reconstructed hydro-climatic temporal series. The
deficiency of observations led to the exploration of various natural archives for
reconstructing past hydro-climatic conditions. Long hydro-climatic series based on natural
proxies in eastern Canada are rare, limited to a tree ring (Boucher et al., 2017; Begin et al.,
2015; Naulier et al., 2015; Nicault et al., 2014; Boucher et al., 2011; Begin et al., 2007;
D'Arrigo et al., 2003) and pollen datasets (Viau et al., 2009) and mainly focused on
temperature reconstructions. Reconstructing river hydrological series using dendrological
analysis is complex in the boreal region due to the indirect relation between tree-ring
indicators and streamflow. One study has reconstructed streamflow variations over the last
two centuries in Labrador based on tree-ring isotopes series (Dinis et al., 2019). Still, the
spatial coverage of palaeohydrological records from independent proxies must be increased
in this region. In this perspective, annually laminated sediments composed of minerogenic
particles (clastic varves) formed when seasonal runoff carrying suspended sediment enters
a lake (Sturm, 1979) have the potential to produce long paleohydrological series. The direct
relationship between clastic varves and hydrological conditions makes this type of varve a
specific and powerful proxy for streamflow reconstructions. Clastic varves can provide, in
favourable settings, annually to seasonally resolved information about downstream
sediment transport from catchment area into lake basin depending on regional hydro-
climatic conditions (Lamoureux, 2000; Lamoureux et al., 2006; Tomkins et al., 2010;
Cuven et al., 2011; Kaufman et al., 2011; Schillereff et al., 2014; Amann et al., 2015;
Heideman et al., 2015; Zolitschka et al., 2015; Saarni et al., 2016; Czymzik et al., 2018).

Preliminary analysis of short sediment cores collected in Grand Lake, central Labrador,
revealed that this lake is an excellent candidate for the preservation of recent fluvial clastic
laminated sediment record (Zolitschka et al., 2015). The objectives of this paper are to: (1)
Confirm the annual character of the laminations record; (2) Establish the relation between
the physical parameters of laminations and local hydro-climatic conditions to examine the
potential proxy for hydrological reconstructions; (3) Reconstruct the hydrology of the last
160 years and compare its similarities and differences with Brigode et al. (2016) rainfall-
runoff modelling over the 1880-2011 period; and (4) Determine if there is a Labrador
regional streamflow signal recorded in Grand Lake laminated sediments.

**2. Regional setting**
Grand Lake is a 245-m-deep (Trottier et al., 2020) elongated (60-km-long) fjord-lake
located in a valley connected to the Lake Melville graben in central Labrador
(53°41'25.58"N, 60°32'6.53"O, ~15 m above sea level) (Fig. 1). The region is part of the
Grenville structural province and is dominated by Precambrian granite, gneiss and acidic
intrusive rocks. Grand Lake watershed deglaciation began after ~8.2 cal ka BP (Trottier et
al., 2020). During deglaciation, marine limit reached an elevation of 120-150 m above
modern sea level and invaded further upstream in the modern fluvial valleys that are
connected to the lake (Fizthugh, 1973). This former glaciomarine/marine sedimentary fjord
basin has been glacio-isostatically uplifted and isolated by a morainic sill to become a deep
fjord-lake (Trottier et al., 2020). The regional geomorphology is characterized by glacially
sculpted bedrock exposures, glacial deposits consisting of till plateaus of various
elevations, glacial lineations, drumlins, kames, eskers and raised beaches (Fulton 1992).
Podzolic soils dominate, with inclusions of brunisols and wetlands.

Grand Lake is located in the High Boreal Forest ecoregion, one of the most temperate
climates in Labrador, hosting mixed forests dominated by productive, closed stands of
Abies balsamea, Picea mariana, Betula papyrifera, and Populus tremuloides (Riley et al.,
2013). This region is influenced by temperate continental (westerly and southwesterly
winds) and maritime (Labrador Current) conditions with cool humid summers (JJA) (~8.5
°C) and cold winters (DJFM) (~-13 °C). The Grand Lake watershed extends upstream over
the low subarctic Nipishish-Goose ecoregion, a broad bedrock plateau (<700 m.a.s.l.)
located on the west flank of the Lake Melville lowlands. Lichen-rich Picea woodlands with
open canopies predominate. With cooler summers and longer cold winters, this area is
slightly influenced by the Labrador Sea. Mean annual precipitation in the study region
ranges from 800 mm to 1 000 mm, with 400 cm to 500 cm of snowfall. The regional
hydrological regime typically exhibits winter low flow and spring freshet, followed by
summer flow recession (Fig. 2). Snowmelt in Grand Lake region takes place from April to
June (AMJ).

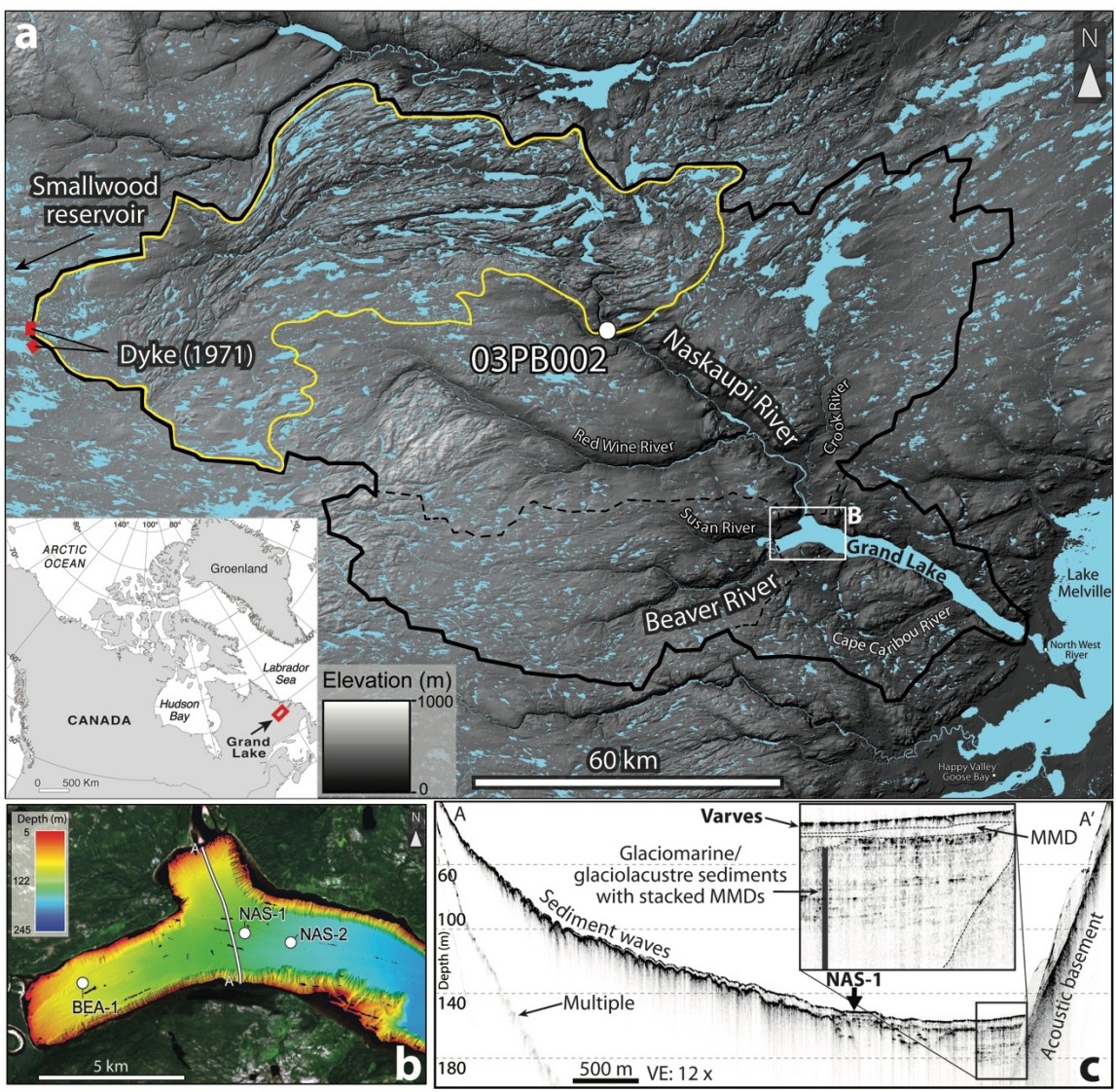

*Figure 1. (A) Location of Grand Lake watershed (black line) and its principal tributaries. The Naskaupi River hydrometric station (03PB002: white dot) covering an area of 4480 km² (yellow line). Location of the dykes constructed in 1971 to divert water from the Naskaupi River to the Smallwood reservoir hydroelectric system are also shown by the red bars. (B) High-resolution swath bathymetry (1-m resolution) of Grand Lake (Trottier et al., 2020) coupled with a Landsat image (USGS) and core site locations. The white line indicates the location of a typical 3.5 kHz subbottom profile (C) of the Naskaupi River delta (A-A') showing the approximate location of core NAS-1.*

The main tributary of Grand Lake is the Naskaupi River located at the lake head (Fig. 1a). The downstream part of the Naskaupi River is fed by the Red Wine and the Crook rivers. The Beaver River is the secondary tributary of Grand Lake. Naskaupi and Beaver rivers structural valleys that connect to the Grand Lake Basin have a well-developed fluvial plain and a generally sinuous course that remobilize former deltaic systems and terraces composed of glaciomarine, marine, fluvio-glacial, lacustrine and modern fluvial deposits.

Upstream river terraces show mass movement scarps and are affected by gully and aeolian
activity. Grand Lake flows into a small tidal lake (Little Lake) and subsequently towards
Lake Melville. On 28 April 1971, by closing a system of dykes, the headwaters of Naskaupi
River watershed (Lake Michikamau) were diverted into the Churchill River hydropower
development (Fig. 1a). This diversion has reduced the drainage area of the Naskaupi River
from 23 310 km$^2$ to 12 691 km$^2$ (Anderson, 1985).

Hydroacoustic data were collected in Grand Lake in 2016 (Trottier et al., 2020). The swath
bathymetric imagery and 3.5 kHz subbottom profile show that the prodelta slopes present
well-defined sediment waves at the Naskaupi River mouth (Trottier et al., 2020; Fig. 1b).
The upper acoustic unit is composed of a high amplitude acoustic surface changing into
low amplitude acoustic parallel reflections (Fig. 1c), a type of acoustic facies which can be
associated with successive sedimentary layers of contrasting particle sizes (Gilbert and
Desloges, 2012).

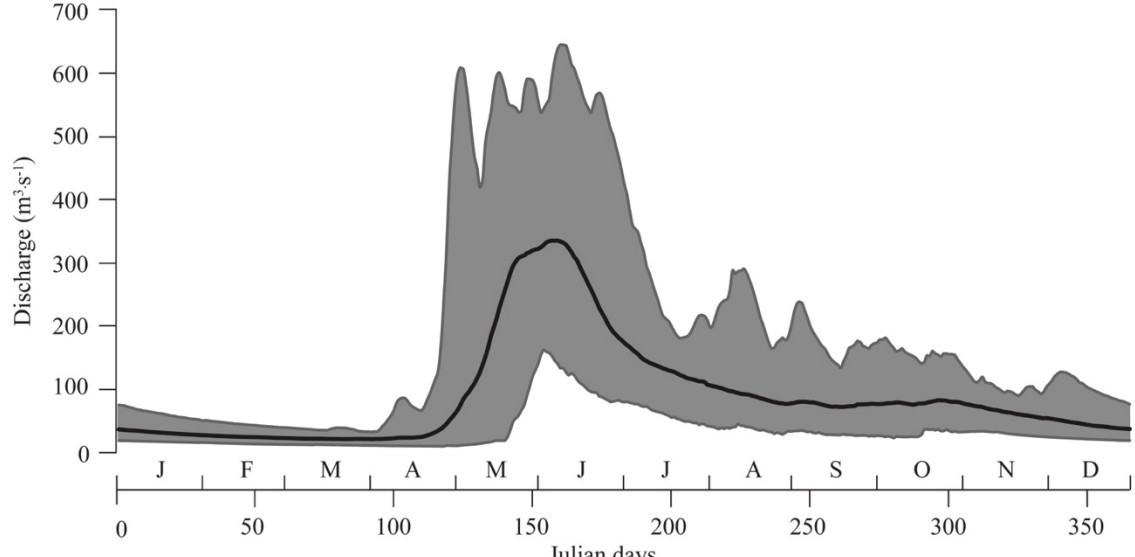


*Figure 2. Observed mean daily discharges of the Naskaupi River (hydrometric station 03PB002) for the*
*1978-2012 period (black line). The gray zone represents the minimum and maximum observed discharges.*

## 3. Methods

### 3.1 Sediment coring and processing



Four short sediment cores (BEA-1, NAS-1A, NAS-1B and NAS-2) were collected using a
UWITEC percussion corer in March 2017 deployed from the lake ice cover. These cores
were collected in undisturbed areas according to the swath bathymetry and subbottom
profiling data (Trottier et al., 2020). Core BEA-1 was collected in the axis of the Beaver
River at a depth of 93 m. Core NAS-1 and NAS-2 were collected in the axis of the Naskaupi
River at a depth of 146 and 176 m, respectively (Fig. 1b). Site BEA-1 and NAS-1 are
located at the distal frontal slope of the Beaver and Naskaupi river deltas (fig. 1c); site
NAS-2 is located away from the Naskaupi River delta, at the beginning of the deep lake
basin. Duplicate cores of different lengths have been retrieved at each site to maximize
undisturbed sediment recovery. Following the extraction of each core, wet floral foam was
gently inserted through the top of the filled coring tube and slowly pushed towards the
sediment surface to seal and preserve the sediment-water interface. A plastic cap was then
installed on top of the foam to secure its position in contact with the intact sediment surface
and avoid disturbance during transport of the cores. The cores were scanned using a
Siemens SOMATOM Definition AS+ 128 medical CT-Scanner at the multidisciplinary
laboratory of CT-scan for non-medical use of the Institut National de la Recherche
Scientifique - Eau Terre Environnement (INRS-ETE). The CT-scan images allowed the
identification of sedimentary structures (i.e., laminated facies, perturbation and hiatus).
Expressed as CT-numbers or Hounsfield units (HU), X-Ray attenuation is a function of
density and the effective atomic number, and hence sensitive to contrasts in mineralogy,
grain size and sediment porosity (St-Onge et al., 2007). CT-numbers were extracted at a
resolution of 0.06 cm using the ImageJ software 2.0.0 (imagej.net). The cores were then
opened, described and photographed with a high-resolution line-scan camera mounted on
an ITRAX core scanner (RGB colour images; 50 µm-pixel size) at INRS-ETE.
Geochemical non-destructive X-Ray Fluorescence (XRF) analysis was performed on the
core half (30 kV and 30 mA). XRF elements profiles were used to visualize the structures
and boundaries of the laminations and estimate particle size variability in sediment cores
(Kylander et al., 2011; Cuven et al., 2010; Croudace et al., 2006). Elements were
normalized by the total of count (cps) for each spectrum. Continuous XRF measurements
were also carried out on overlapping impregnated sediment blocks in order to superpose
element relative intensity profiles on thin-sections.

**3.2 Chronology and thickness measurement**

Surface sediments from cores BEA-1 and NAS-1A were dated with $^{137}$Cs method (Appleby
and Oldfield 1978) using a high-resolution germanium diode gamma detector and
multichannel analyzer gamma counter. $^{137}$Cs activity was used to identify sediment
deposited during 1963-1964 peak of nuclear tests and validate the annual character of the
layers. A sampling interval of 2 cm was used to approximately identify the depth at which
the $^{137}$Cs peaks were located. Subsequently, a sampling interval of ± 0.5 cm was used to
sample each lamination for the period 1961-1965 to determine the exact $^{137}$Cs peak location
(1963-1964). In order to establish a chronology for each core, detailed laminations counts
were executed on CT-scan images and high-resolution photographs using ImageJ 2.0.0 and
Adobe Illustrator CC softwares (Francus et al., 2002). As all of the core surface has been
well preserved, the first complete lamination below the sediment surface was considered
to represent the topmost year (i.e., 2016 CE). Chronology on each core was confirmed by
cross-correlation between thick laminations selected as distinctive marker layers along the
different sediment sequences (A to M; Fig. 4).

Thin-sections of sediments were sampled from cores BEA-1 (1856-2016), NAS-1A (1953-
2016), NAS-1B (1856-1952) and NAS-2 (1968-2016) (see Fig. 4 for thin-section location)
following Francus and Asikainen (2001) and Lamoureux (1994). Digital images of the thin-
sections were obtained using a transparency flatbed scanner at 2400 dpi resolution (1 pixel
= 10.6 µm) in plain light and were used to characterize lamination substructure. Lamination
counts and thickness measurements using a thin-section image analysis software developed
at INRS-ETE (Francus and Nobert 2007) were performed to duplicate and validate
previous chronologies established on CT-Scan images and high-resolution photographs.
Two counts were made from thin-section by the same observer (AGP). Total Varve
Thickness (TVT) and Detrital Layer Thickness (DLT) of each year of sedimentation were
measured from images of thin-sections. Lamination counts made on CT-scan images, high-
resolution photographs and thin-sections are identical while TVT measurements show
negligible difference ($R^2$ = 0.96; p < 0.05). The thickness measurements made from CT-
scan images and high-resolution photographs have been used to prolong the TVT series of
core NAS-2 from 1968 back to 1856. Continuous TVT measurements allowed the
establishment of high-resolution age-depth models for each site.
**3.3 Image and particle size analysis**
Using custom-made Image Analysis software (Francus and Nobert, 2007), regions of
interest (ROIs) were selected on the thin-section images. The software then automatically
yielded SEM images of the ROIs using a Zeiss Evo 50 scanning electron microscope
(SEM) in backscattered electron (BSE) mode. Eight-bit greyscale BSE images with a
resolution of 1024 x 768 pixels were obtained with an accelerating voltage of 20 kV, a tilt
angle of 6.1 and an 8.5 mm working distance with a pixel size of 1 µm. BSE images were
processed to obtain black and white images where clastic grains (>3.5 µm) and clay matrix
appeared black and white respectively (Francus, 1998).

Each sedimentary particle (an average of 2 225 particles per image) was measured
according to the methodology used by Lapointe et al. (2012), Francus et al. (2002) and
Francus and Karabanov (2000) in order to calculate particle size distribution on each ROI
image. Due to the thickness of the laminations, results from several ROI images were
merged to obtain measurements for each year of sedimentation, with an average of 4
images per lamination. Only clastic facies related to spring and summer discharges were
used for particle size analysis in order to exclude ice-rafted debris (µm to mm scale)
observed in the early spring layers (see Fig. 5 for details). The 99th percentile (P99D$_0$) of
the particle size distribution for each detrital layer was obtained from thin-sections
(Francus, 1998) for the last 160 years (1856-2016) for core BEA-1 and NAS-1, and for the
last 47 years (1968-2016) for core NAS-2, from 795, 717 and 132 BSE images respectively
(Fig. 4).

**3.4 Hydro-climatic variables**
Hydrological variables (Tab.1) were calculated from the time series of daily discharges
recorded by the Naskaupi River hydrometric station over the 1978-2011 period (missing
data from the years 1996, 1997 and 1998).

*Table 1. Hydro-climatic variables used in this paper*

| Hydrological variable | Unit | Description |
|---|---|---|
| Q-max | m³/s | Annual maximum of daily discharges |
| Q-mean | m³/s | Mean annual discharge |
| Q-max-Jd | Julian days | Julian day at which the discharge reaches its maximum annual value |
| Rise-Time | Days | Number of days between the minimum winter flow and the maximum spring flow |
| Nb-Days-SupQ80 | Days | Number of days with discharge greater than the $80^{th}$ daily percentile |
| Q-nival | mm | Nival runoff (April, May, June, July) |
| Snow-Win | mm | Winter snowfall (September to May) |
| Ptot-Annual | mm | Winter Snowfall + Summer rainfall |
| Ptot-Summ | mm | Summer rainfall (March to October) |
| Temp-Spring | °C | Average spring temperature (April, May, June) |



The Naskaupi River hydrological variables have been compared with four other
hydrometric station data available around the study region (Fig. 3a, Tab. 2), which are
devoid of anthropogenic perturbations. Q-mean series from the five stations have been
normalized for the common 1979–2011 period and averaged, to produce a Labrador region
mean annual discharge series. This allows to extend instrumental data series for the period
1969 to 2011, and fill in data for the missing years. The Labrador hydrometric station data
used in this study come from a Government of Canada website (https://wateroffice.ec.gc.ca

268 05/2018).


Table 2. Description of hydrometric stations used in this study

| Hydrometric station | ID | Area (km²) | Location (N,W) | Recording period |
|---|---|---|---|---|
| Ugjoktok River | 03NF001 | 7570 | 55° 14' 02", 61° 18' 06" | 1979-2011 |
| Naskaupi River | 03PB002 | 4480 | 54° 07' 54", 61° 25' 36" | 1978-2011 |
| Minipi River | 03OE003 | 2330 | 52° 36' 45", 61° 11' 07" | 1979-2011 |
| Little Mecatina River | 02XA003 | 4540 | 52° 13' 47", 61° 19' 01" | 1979-2011 |
| Eagle River | 03QC001 | 10 900 | 53° 32' 03", 57° 29' 37" | 1969-2011 |


**3.5 Varve physical parameters and hydrological variables**

A simple linear regression model was used to fit the DLT and $P99D_0$ series with local (1978-2011) and regional (1969–2011) instrumental series and reconstructed hydrological variables (Q-mean, Q-max) back to 1856. Model calibration was performed using a twofold cross-validation technique over the instrumental period. Root mean squared errors (RMSE) and coefficient of determination ($R^2$) were calculated for calibration periods, while average reduction of error (RE) and average coefficient of efficiency (CE) were calculated to evaluate reconstruction skills (Briffa et al. 1988, Cook et al., 1999). The RE and CE of the verification periods must be > 0 to validate the model skills. Statistical analysis was realized using the treeclim package (Zang and Biondi, 2015) in the R-project environment (R Core Team, 2019, http://www.r-project.org/).

**3.6 Hydro-climatic reconstruction based on rainfall-runoff modelling**

The applied reconstruction method is based on rainfall-runoff modelling. Firstly, it aims at producing, for the Naskaupi River hydrometric station catchment (Fig. 1a), daily climatic time series using a historical reanalysis of global geopotential height fields extracted over the studied region for a given time period (here 1880-2011). Secondly, the produced climatic series are used as inputs to a rainfall–runoff model previously calibrated on the studied catchment in order to obtain daily streamflow time series. The reconstruction method is fully described in Brigode et al. (2016) and was recently applied over southeastern Canada catchments in Dinis et al. (2019). It is summarized in the following paragraphs.

The available observed hydro-climatic series for the Naskaupi River hydrometric station catchment have been aggregated at the catchment scale. Climatic series (daily air temperature and precipitation) have been extracted from the CANOPEX dataset (Arsenault et al., 2016), built using Environment Canada weather stations and Thiessen polygons to calculate climatic series at the catchment scale. Daily air temperature series have been used for calculating daily potential evapotranspiration at the catchment scale, using the Oudin et al. (2005) formula designed for rainfall-runoff modelling.

These daily series have been used for calibrating the GR4J rainfall-runoff model (Perrin et
al., 2003) and its snow accumulation and melting module, CemaNeige (Valéry et al.,
2014a), using the airGR package (Coron et al., 2017). This combination of GR4J and
CemaNeige (hereafter denoted CemaNeigeGR4J) has been recently applied over eastern
Canada catchments and showed good modelling performances (e.g., Seiller et al., 2012;
Valéry et al., 2014b, Brigode et al., 2016). CemaNeigeGR4J has been calibrated on the
recorded period of the Naskaupi River hydrometric station catchment using the Kling and
Gupta efficiency criterion (Gupta et al., 2009) as objective function.

Then, the observed climatic series have been resampled over the 1880-2011 period, based
on both season and similarity of geopotential height fields (Kuentz et al., 2015). The
resampling is performed by calculating Teweles and Wobus (1954) distances between four
geopotential height fields: (i) 1000 hPa at 0 h, (ii) 1000 hPa at 24 h, (iii) 500 hPa at 0 h,
and (iv) 500 hPa at 24 h. The NOAA 20[th] Century Reanalysis ensemble (Compo et al.,
2011, hereafter denoted 20CR) has been used as a source of geopotential height fields (Fig.
3b).

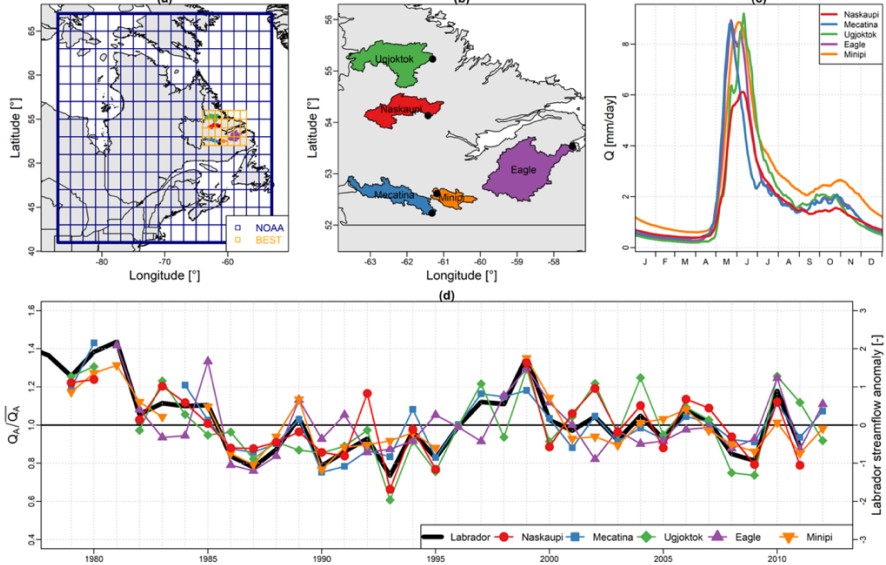


*Figure 3. (a) Dataset used for the hydro-climatic reconstruction based on rainfall-runoff modelling: the*
*extension of the 20CR grid used is shown in blue, while the BEST grid used is highlighted in orange. (b)*
*Spatial distribution of hydrometric stations used in this study (black dots) and their catchment area. (c)*
*Observed mean daily discharges of each hydrometric station for the 1978-2012 period. (d) Labrador*
*streamflow anomaly and **the Labrador region** mean annual discharge series (thick black line).*
As in Brigode et al. (2016), the resampled series of air temperature have been corrected at
the catchment scale using a regression model calibrated with the Berkeley Earth Surface
Temperature analysis (Rohde et al., 2013, hereafter denoted BEST). BEST is a gridded air
temperature product starting in 1880 at the daily timestep (Fig. 3b).

Finally, the daily climatic series are used as inputs to the CemaNeigeGR4J model in order
to obtain daily streamflow time series on the same 1880-2011 period. Thus, the outputs of
the hydro-climatic reconstruction are an ensemble of daily meteorological series (air
temperature, potential evapotranspiration and precipitation) and an ensemble of daily
streamflow series.

**4. Results**
**4.1 Lamination characterization**
Sediment retrieved at the head of Grand Lake (Fig. 4), consist of dark grayish to dark
yellowish brown (Munsell colour: 10YR-4/2 to 10YR-4/4) laminated minerogenic
material, interpreted as clastic lamination of fluvial origin. Lamination structure can be
divided in 3 seasonal layers (Fig. 5) based on their stratigraphic position and microfacies.
Annual sedimentation starts with a layer composed of silt and clay sediment matrix which
sometimes contains ice-rafted debris (μm to mm scale) interpreted as an early spring layer.
The major lamination component is a spring and summer/autumn detrital layer. Its thick
basal part is mostly poorly sorted, graded and composed of coarse minerogenic grains
comprising fine sand and silts (< 150 μm) with some redeposited cohesive sediment clasts
eroded from the underlying early spring layer. This detrital layer has a sharp lower
boundary. The upper part of the detrital layer consists of a finer detrital grain matrix
containing thin visually coarser intercalated sub-layers in ~75% of the laminations. The
allochthonous lithoclastic materials which compose the detrital layers are associated with
higher density values (Fig. 4) and an increase in the relative intensity of elements Sr and
Ca (Zolitschka et al., 2015). Few organic debris and charcoal fragments are observed
throughout the detrital layers. The third topmost lamination layer is formed by a fine to
medium silty layer with abundant clay rich in Fe and interpreted as an autumn and winter
layer, also known as a clay cap (Zolitschka et al., 2015). The Fe peak values in autumn and

winter layers, are hence used to determine the upper lamination boundary (Fig. 4) (Zolitschka et al., 2015) as previously performed in other varved sequences (Cuven et al., 2010; Saarni et al., 2016).

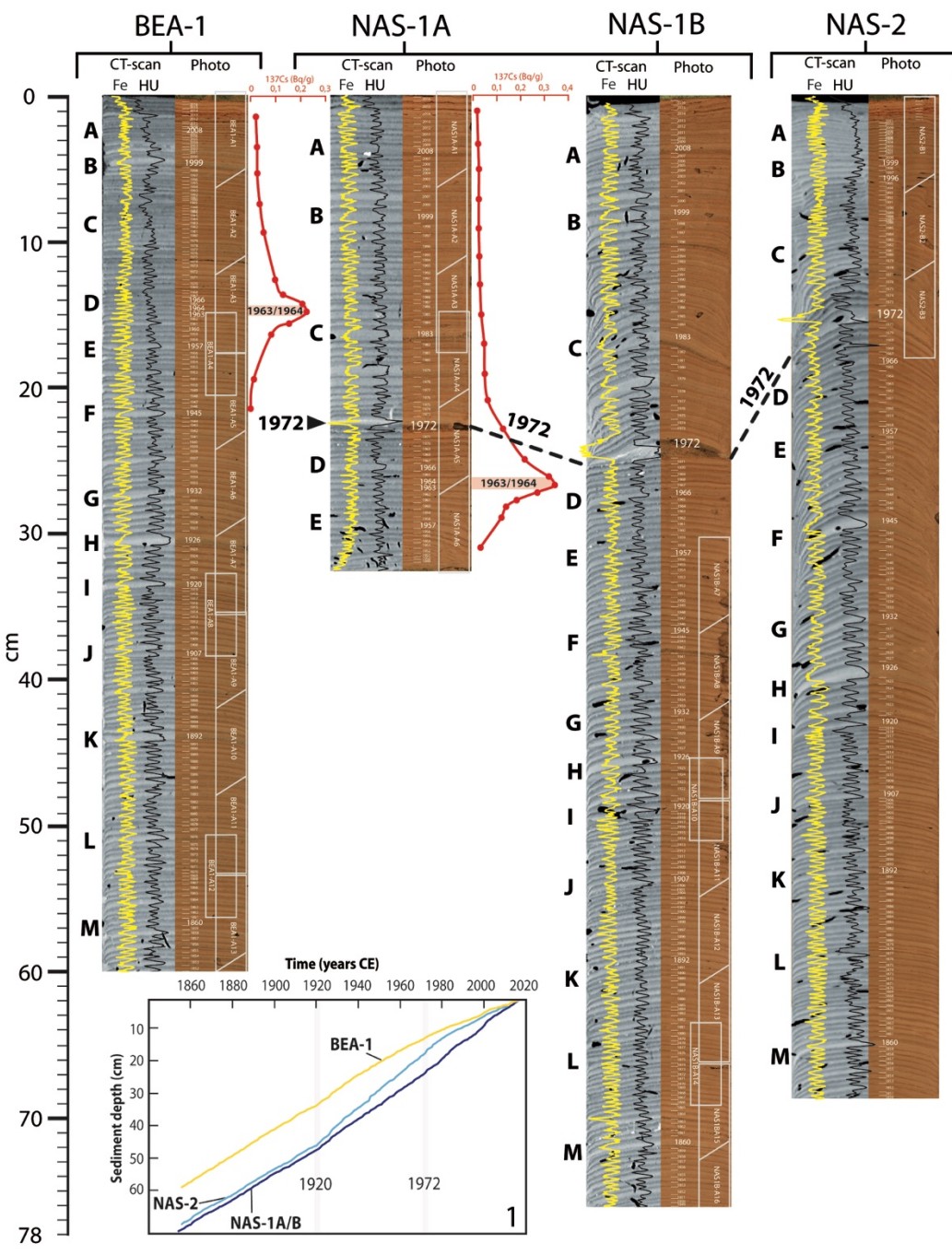

*Figure 4. Varve counts made on (left) CT-scan and (right) high resolution images from core BEA-1, NAS-1A/B and NAS-2. Distinctive marker layers are identified by letters A to M. The 1972 marker layer is outlined by the thick dark gray line. Fe relative intensity and density (HU) profile represented by the yellow and black line respectively, show rhythmic laminations. The activity profile of $^{137}$Cs in core BEA-1, NAS-1A is shown by the red line. Approximate thin-section locations are outlined by white boxes. The age-depth model of the 3 cores is also presented (Box. 1). See Fig. 1b for core locations.*

The lamination deposited in 1972 from sites in the axis of the Naskaupi River (NAS-1; Fig. 5b and NAS-2; Fig. 4), present a thick (8.2 mm) and coarse (67.8 µm) detrital layer composed of very fine sandy and very coarse silt (Fig. 5b) representing the highest particle size measured in all sequences. Furthermore, there is a difference in lamination physical parameters and microfacies deposited before and after the 1972 marker bed, especially in core NAS-1, the proximal site from the Naskaupi River mouth. Laminations deposited prior 1972 have a well-developed substructure relatively constant among each annual lamination (Fig. 5b). The early spring layer of the pre-1972 laminations is thicker and more clearly visible. Conversely, the detrital layer of laminations post-1972 is thicker, while the early spring layer is more difficult to discern and contributes less to the TVT (Fig. 5a). The mean contribution of the early spring layer and autumn and winter layer to the total lamination thickness is 35% for the pre- and 52% for the post-1972 intervals. The early spring layer in lamination post-1971 from sites NAS-1 and NAS-2 no longer contains isolated coarse debris. The changes in lamination facies are less noticeable in core NAS-2, which was sampled further away from the Naskaupi River mouth. The 1972 marker bed and related facies changes are not found at the Beaver River mouth site BEA-1.

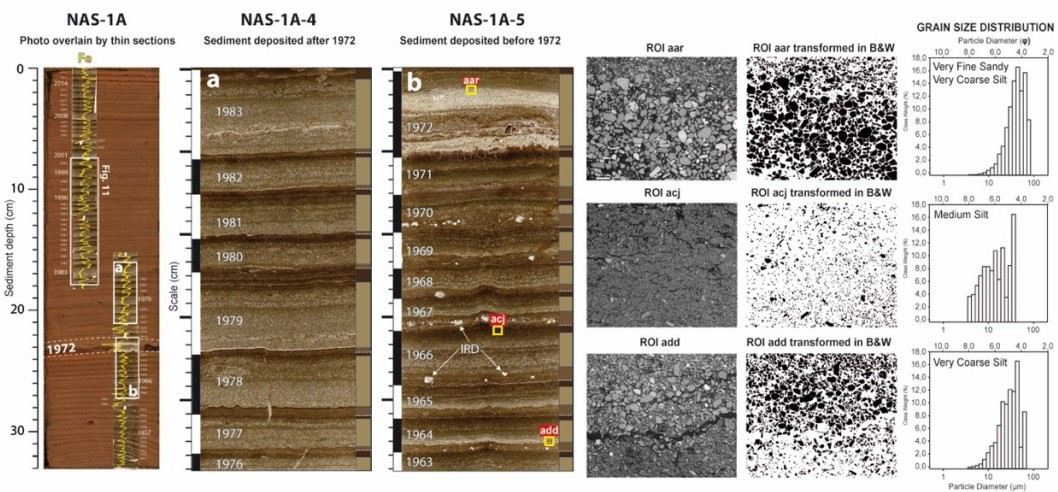

*Figure 5. (Left) Photo of core NAS-1A overlain by thin-section image and Fe relative intensity profile (yellow lines). The 1972 marker layer is outlined by the white dashed lines. Thin-section images showing sedimentary structure of varves deposited (B) before and (A) after the 1972 marker bed. Varve boundaries are represented by the vertical black and white bars. Varve layers are delimited by the medium brown (early spring layer), pale brown (detrital layer) and dark brown (autumn and winter layer) bars. Typical Ice-Rafted Debris (IRD) are shown by the white arrows on the b panel. (Right) BSE images of three ROIs transformed in B&W and their associated particle size distribution (aar: the 1972 marker layer; acj: a typical autumn and winter layer; add: the base of a typical detrital layer) (see yellow squares on the b panel for ROIs location).*

**4.2 Varve chronology**

The laminated sequences chronologies are consistent with the Cesium-137 main peaks corresponding to the highest atmospheric nuclear testing period (1963-1964 CE) (Appleby, 2001). Peaks are found at 14-14.5 cm (BEA-1) and 26.5-27 cm (NAS-1A) depth (Fig. 4) and perfectly match the lamination counts in both cores, confirming the varve assumption. The presence of the distinct 1972 marker layer at this chronostratigraphic position in the varve sequence which coincides with the occurrence of the Naskaupi River diversion that took place in April 1971 (see section 5.2 for details) supports the reliability of the constructed chronologies.

Independent varve chronologies were established from sediment cores BEA-1, NAS-1 and NAS-2 (Fig. 4). A total of 160 varves were counted at each site, covering the 1856-2016 period. The thickness and the good quality of the well-preserved varve structures allowed a robust age-model reproducible among cores to be constructed. Despite the distance between the coring sites (1 to 5 km) and the two different sediment sources (Naskaupi and Beaver River) (Fig. 1b), there is no varve count difference between the selected thick marker layers (A to M; Fig. 4) among cores. The few counting difficulties occur within varve years 1952-1953, 1935-1934, 1918-1919, as it contains ambiguous coarse non-annual intercalated sub-layers with intermediate clay cap that can be interpreted as one year of sedimentation. Both varve counts performed on thin-sections show a low overall counting error (±1.8%) which demonstrated the precision and accuracy of the varve sequences chronology. The age-depth models (Fig. 4, Box. 1) show changes in sediment accumulation rates (thickness) among cores in 1920 and 1972.

**4.3 Thickness and particle size measurements**

The TVTs from core BEA-1, NAS-1 and NAS-2 vary between 0.9 and 12.9 mm, with an average thickness of 4.09 mm (Fig. 6a, b, c, Supplements Fig. S1 and Tab. S1). The DLTs vary between 0.3 and 8.3 mm, with an average thickness of 1.9 mm (Fig. 6a, b, c, Supplements Fig. S2 and Tab. S2). There are significant strong positive correlations between TVT and DLT for each core (r = 0.79 to 0.91; p < 0.01). A step in the TVT is observable in the early 1920s at the three sites (Fig. 6a, b, c), especially in core NAS-2,

which recorded their highest values (12.9 mm) during the 1920-1972 period (Fig. 6c).
Since the 1920s, there is a statistically significant decreasing trend in TVTs and DLTs in
core BEA-1 (Fig. 6a). Thickness data from the three sites have been normalized and
averaged to produce combined TVT and DLT series (Fig. 6d, e). From 1920 to 1972,
combined TVT and DLT series show a statistically significant downward trend, despite an
increase in years associated with high thickness values. Overall, TVT and DLT vary
similarly in time between sites during the 1856-1971 period (Fig. 6d, e). However, after
1972, TVT and DLT series are more diverging. From 1972 to 2016, there is a statistically
significant decreasing trend in TVT and DLT in cores NAS-2 (Fig. 6c), and the amplitude
of their variability tends to diminish. For core NAS-1 (Fig. 6b), post-1971 period is
associated with higher thickness values. Core NAS-1 has recorded a slight TVT and DLT
decrease for the 1972-2016 period, but unlike the other cores, the variability tends to
increase. The TVT and DLT are overall finer in the distal core NAS-2 compared to the
more proximal core NAS-1 (Fig. 4, Box. 1, Supplements Tab. S1, S2).


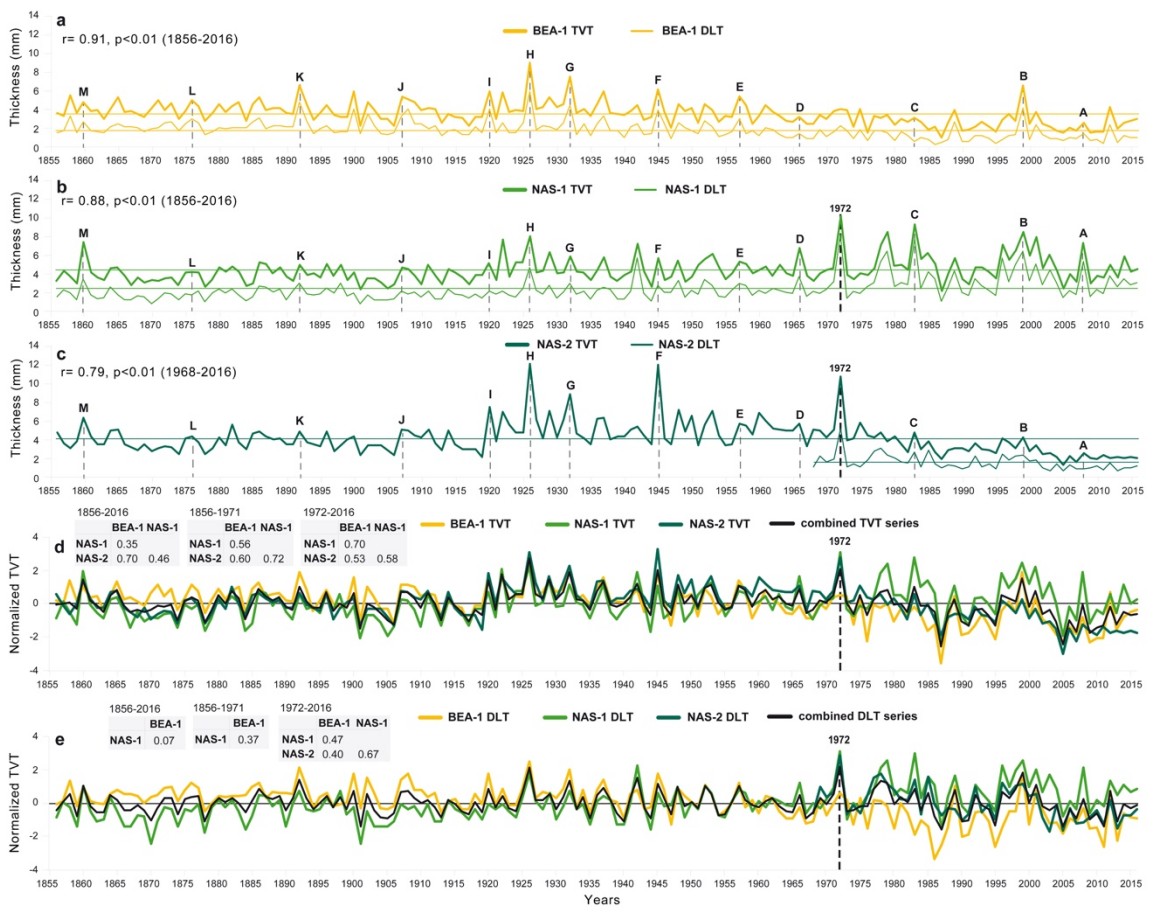

*Figure 6. Total Varve Thickness (TVT; thick line) and Detrital Layer Thickness (DLT; thin line) time series*
*of core (a) BEA-1, (b) NAS-1 and (c) NAS-2. Normalized (d) TVT and (e) DLT series and the combined series*
*(mean of the normalized data from the 3 sites). Pearson correlation coefficients between TVT and DLT for*
*the 1856-2016, 1856-1971 and 1973-2016 periods are shown. The selected marker layers are identified by*
*letters A to M and the 1972 marker layer is outlined by the thick black dashed line.*
The P99D$_0$ of cores BEA-1, NAS-1 and NAS-2 vary between 20 and 67.8 µm, with an
average value of 34.3 µm (Fig. 7, Supplements Fig. S3 and Tab. S3). The grain size is finer
in core NAS-2 compared to core NAS-1. Particle size data from the three sites have been
normalized and averaged to produce combined P99D$_0$ series (Fig. 7c). The combined
P99D$_0$ series show a slight coarsening trend towards the end of the 19$^{th}$ century. From 1900
to 1971, P99D$_0$ values are generally below average. The 1972 marker layer of core NAS-
1 presented the maximum P99D$_0$ values (Fig. 7b). After 1972, there is an increase of P99D$_0$
values in core NAS-1, where a step is observable. Pre-1971 varves in core NAS-1 have a
mean P99D$_0$ of 32,47 µm compared to 42,91 µm for the 1972-2016 period.

There is weak to moderate positive correlation between TVT and $P99D_0$ from a same core
(BEA-1: r = 0.41 p < 0.01; NAS-1: r = 0.52 p < 0.01; NAS-2: r = 0.27, p < 0.05). The
correlation between DLT with $P99D_0$ is stronger (BEA-1: r = 0.49 p < 0.01; NAS-1: r =
0.65 p < 0.01; NAS-2: r = 0.49, p < 0.01). Thick varves are more likely to have high grain
size values. However, these correlations show that TVT, DLT and $P99D_0$ remain
independent variables and can both reveal different hydrological information.

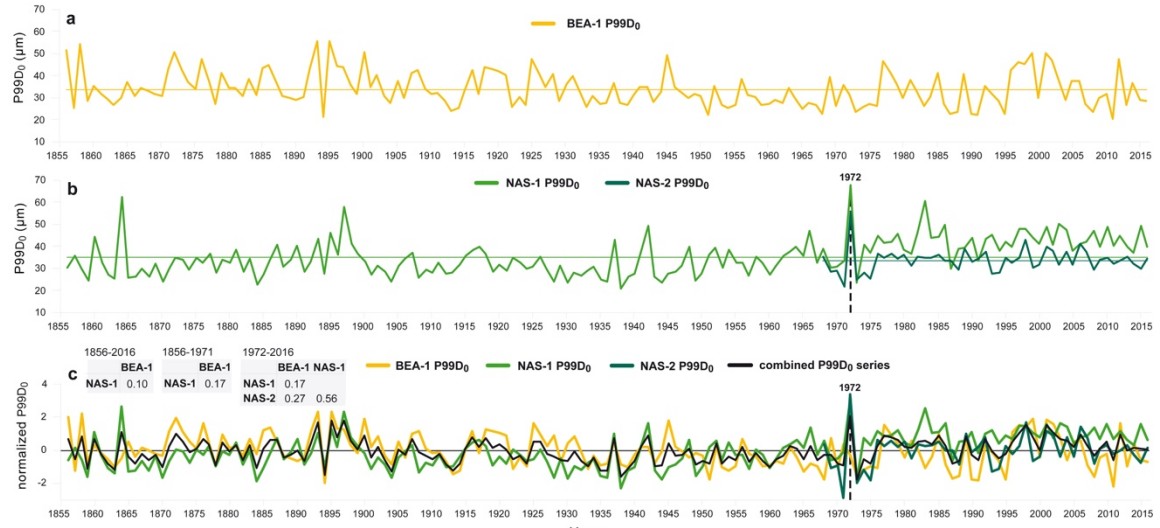

*Figure 7. $P99D_0$ time series of cores (a) BEA-1, (b) NAS-1 (1856-2016) and NAS-2 (1968-2016). (c)*
*Normalized $P99D_0$ series and the combined series (mean of the normalized data from the 3 sites). The 1972*
*marker layer is outlined by the black dashed line. Pearson correlation coefficients between $P99D_0$ series for*
*the 1856-2016 and 1968-2016 periods are shown.*
**4.5 Relation between varve series and instrumental record**
*4.5.1 Naskaupi River*
To examine how the physical parameters of the varves are related to local hydroclimate
and to demonstrate their potential for hydrological reconstruction, sediment parameters
(TVT, DLT and $P99D_0$) of each core were systematically compared to hydrological
variables (Tab. 1). TVT, DLT and $P99D_0$ series from the three coring sites show significant
positive correlations with the Q-mean and Q-max extracted from the Naskaupi River
hydrometric station (03PB002) data on the 1978-2011 period (n=31) (Tab. 3). The TVT
and DLT of cores BEA-1 and NAS-2 show stronger correlation with Q-mean, while TVT
and DLT of cores NAS-1 have a better relation with Q-max. There is a significant negative
correlation between $P99D_0$ of core NAS-1 and Q-max-Jd (r = -0.38) and Rise-Time
(r = - 0.47). Sediment parameters also present significant positive correlations with Q-
Nival (r = 0.32 to 0.61), Snow-Win (r = 0.47 to 0.61) and Nb-days-SupQ80 (> 125 $m^3 \cdot s^{-1}$)
(r = 0.44 to 0.62). Moreover, the maximum particle size series of core NAS-1 show
significant (p = 0.02) positive correlations with the average spring temperature (r = 0.40;
not shown in Tab. 3). Combined DLT and $P99D_0$ series (Fig. 6d, e; 7c) yields the strongest
correlations in our dataset (r = 0.68 and 0.75; Tab. 3) and have been used to reconstruct
Naskaupi River Q-mean and Q-max respectively (Fig. 8).

*4.5.2 Labrador region*
To determine if there is a regional hydrological signal in Labrador and whether the Grand
Lake varved sedimentary sequence has recorded this signal, the Naskaupi River hydro-
climatic variables were compared with other Labrador hydrometric stations (Tab. 2).
Despite specific local geomorphological and climatic conditions, strong similarities exist
between observed mean daily discharges (Fig. 3c) and annual streamflow (Fig. 3d)
recorded by hydrometric stations in Labrador for the 1978-2011 period. The shape of the
five annual regimes shows similar characteristics (i.e. flood-timing, strength, duration,
snowmelt and rainfall response). The instrumental Naskaupi River mean annual discharge
series data show significant (p < 0.01, Supplements Tab. S5) positive correlations with
other hydrometric stations (Ugjoktok: r = 0.84; Minipi: r = 0.70; Little Mecatina: r = 0.73;
Eagle: r = 0.49). Hydrological conditions in the Naskaupi river region is thus representative
of a broader region of Labrador. Therefore, the combined DLT series (without the NAS-1
1978-2016 period) has been used to reconstruct the Labrador region mean annual discharge
series (Fig. 9).

*Table 3. Matrix of correlation coefficients (Pearson r) of the hydro-climatic variables defined in Tab. 1 with Total Varve Thickness (TVT), Detrital Layer Thickness (DLT) and particle size (P99D$_0$) on the instrumental period (1978-2011; n=31) for each core. Correlations between the hydro-climatic variables and the combined TVT, DLT and P99D$_0$ series (normalized and averaged varve parameters of cores BEA, NAS-1 and NAS-2) are also present. Correlations in boldface are significant at p < 0.05 (Supplements Tab. S4). Correlations marked by an asterisk were used for the final Q-mean and Q-max reconstructions.*

**Hydroclimatic variables of station 03PB002**

| Core BEA-1 | Q-mean | Q-max | Q-max-Jd | Rise-Time | Nb-days-supQ80 | Q-nival | Snow-Win |
|---|---|---|---|---|---|---|---|
| TVT | **0.53** | **0.46** | -0.19 | -0.06 | **0.54** | **0.41** | **0.47** |
| DLT | **0.54** | **0.38** | -0.01 | 0.22 | **0.44** | **0.32** | 0.29 |
| P99D$_0$ | **0.56** | **0.56** | -0.05 | 0.17 | 0.34 | **0.40** | 0.24 |

| Core NAS-1 | Q-mean | Q-max | Q-max-Jd | Rise-Time | Nb-days-supQ80 | Q-nival | Snow-Win |
|---|---|---|---|---|---|---|---|
| TVT | **0.52** | **0.64** | -0.31 | -0.26 | **0.55** | **0.56** | **0.55** |
| DLT | **0.53** | **0.67** | -0.31 | -0.27 | **0.53** | **0.54** | **0.50** |
| P99D$_0$ | 0.19 | **0.60** | **-0.38** | **-0.47** | 0.26 | **0.40** | 0.30 |

| Core NAS-2 | Q-mean | Q-max | Q-max-Jd | Rise-Time | Nb-days-supQ80 | Q-nival | Snow-Win |
|---|---|---|---|---|---|---|---|
| TVT | **0.49** | **0.45** | 0.04 | -0.24 | **0.56** | **0.47** | **0.61** |
| DLT | **0.62** | **0.57** | 0.07 | -0.13 | **0.59** | **0.61** | **0.60** |
| P99D$_0$ | 0,39 | **0.43** | 0.19 | 0.26 | 0.31 | **0.40** | 0.11 |

| combined series | Q-mean | Q-max | Q-max-Jd | Rise-Time | Nb-days-supQ80 | Q-nival | Snow-Win |
|---|---|---|---|---|---|---|---|
| TVT | **0.56** | **0.58** | -0.19 | -0.20 | **0.60** | **0.53** | **0.59** |
| DLT | **0.68*** | **0.65** | -0.11 | -0.07 | **0.62** | **0.58** | **0.54** |
| P99D$_0$ | **0.59** | **0.75*** | -0.09 | 0.05 | **0.43** | **0.56** | 0.23 |

*(Sediment parameters)*

## 4.6 Hydrological reconstructions using varve parameters

### 4.6.1 Naskaupi River Q-mean and Q-max

The Naskaupi River mean and maximum annual discharges (Q-mean and Q-max) were reconstructed using DLT and P99D$_0$ series for the 1856–2016 period. The reconstructions were performed using single-core data, combined DLT and P99D$_0$ series and other combinations of core data, in order to propose the most relevant reconstructions (Supplements Fig. S4, S5). The observations and the reconstructed Q-mean and Q-max extracted from the different series over the 1978-2011 period are consistent. Despite differences, all reconstructions tested using different sources of sedimentological data generally share common interannual and longer-term variability.

Excluding the 1972-2016 measurements from NAS-1 from the combined series for reconstructions was also tested to remove the likely anthropogenic impact on sedimentation during this period. The combined DLT series without the 1972-2016 period presents a

slightly better fit with the instrumental data (lowest RMSE and the most-significant and
highest $R^2$, Supplements Tab. S6). The model calibrations based on a twofold cross-
validation reveal that this DLT series has better overall predictive capacity to reconstructed
Q-mean (Supplements Tab. S7). The 1972-2016 period of core NAS-1 was then excluded
from the combined DLT series used to perform the best reconstruction of Naskaupi River
Q-mean presented in Fig. 8a. However, significantly stronger calibration and validation
statistical results were obtained by keeping this period in the combined $P99D_0$ series used
to reconstruct Naskaupi River Q-max (Fig. 8b, Supplements Tab. S8, S9). The varve of
year 1972 is considered as an outlier that originated from anthropogenic impacts, and thus
was not included in all reconstructions.

The reconstructed Naskaupi River Q-mean from combined DLT series varies between 73
and 126 $m^3 \cdot s^{-1}$, with an average of 96 $m^3 \cdot s^{-1}$ (Fig. 8a), and remains relatively stable from
1856 to 1920, mainly near average. Several years with high Q-mean occurred during the
1920-1960 period. A statistically significant downward trend of the Q-mean is observed
over the last 90 years. Recently, high Q-mean periods are observed from 1976 to 1985 and
1996 to 2002 and lower Q-mean periods from 1986 to 1995 and 2003 to 2016. The
reconstructed Naskaupi Q-max from combined $P99D_0$ series varies between 192 and 681
$m^3 \cdot s^{-1}$, with an average of 426 $m^3 \cdot s^{-1}$ (Fig. 8b). There is a slight upward trend in Q-max at
the end of the 19th century. The 1900-1971 period is characterized by a Q-max generally
below average. Three periods of high Q-max are observed from 1887 to 1900, 1976 to
1986 and 1995 to 2008 (Fig. 8b).

*4.6.2 Labrador region Q-mean*
The consistency between combined DLT series and the observed Labrador region Q-mean
series (Fig. 9), based on the discharge variability of five watersheds of different size and
location, demonstrates that the Grand Lake varved sequence contains a regional signal. The
best reconstruction of Labrador region mean annual discharges is the one performed using
the combined DLT series without the NAS-1 1972-2016 period. This reconstruction
demonstrates the best predictive capacity (RE and CE must be > 0 to validate the model
skills, Supplements Tab. S10, S11). The regional Q-mean reconstruction for the 1856–
2016 period is presented in Fig. 9.

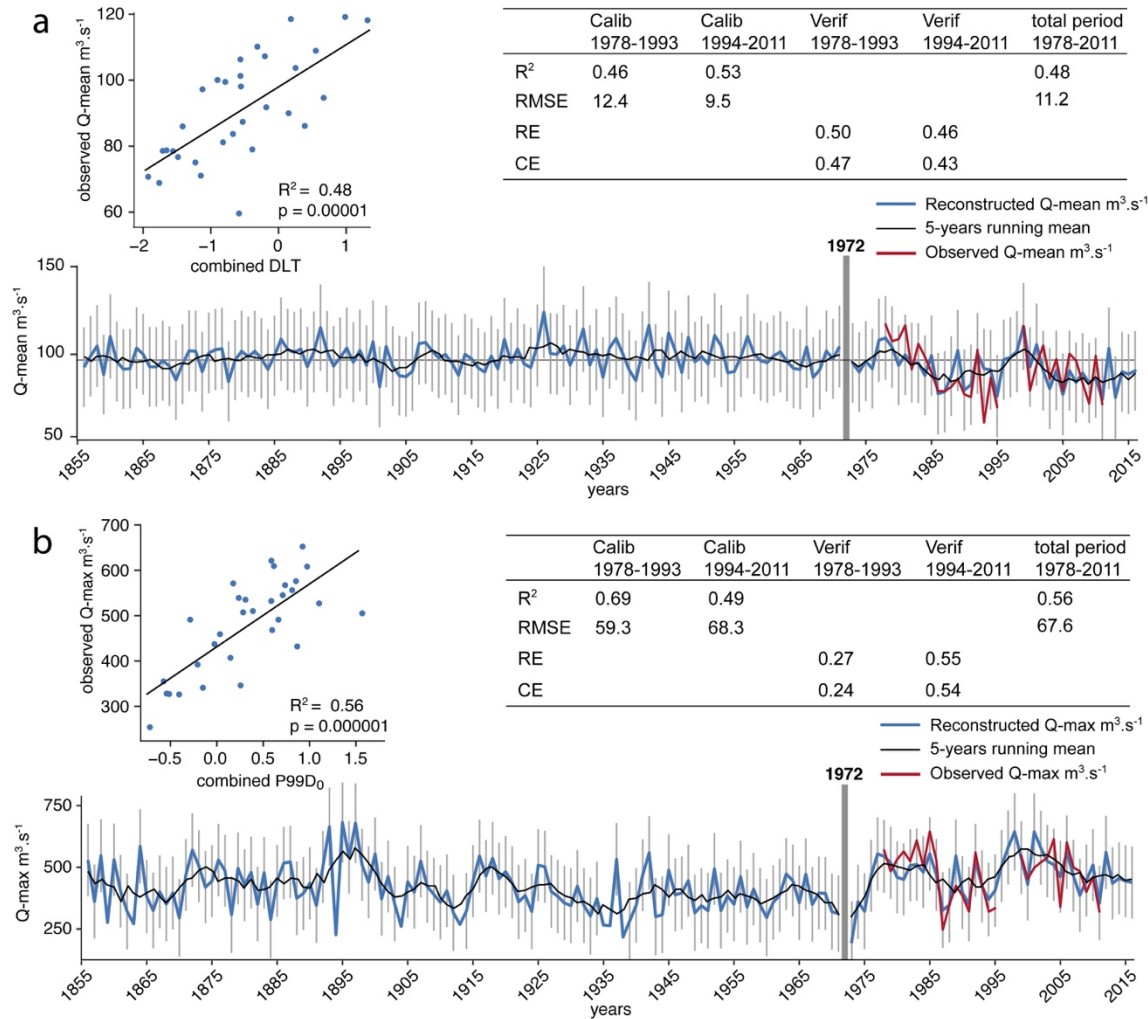

*Figure 8. Naskaupi River (a) Q-mean and (b) Q-max reconstructed from combined DLT (Without the NAS-*
*1 1978-2016 period) and P99D₀ series respectively, for the 1856–2016 period (blue line), with 5-year*
*moving average (black line). Error bars represent the 95% confidence interval. Observed Q-mean and Q-*
*max are also shown for the 1978-2011 period (red line).*

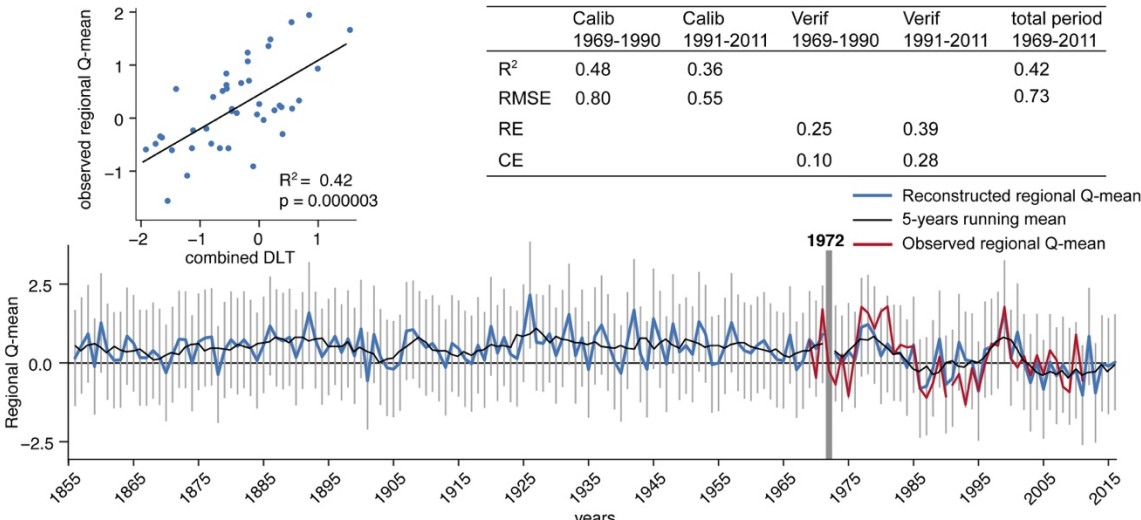

Figure 9. Labrador region Q-mean reconstructed from combined DLT series (without the NAS-1 1972-2016 period) for the 1856–2016 period (blue line), with 5-year moving average (black line). Error bars represent the 95% confidence interval. Observed Labrador region Q-mean series is also shown for the 1969-2011 period (red line).

## 4.7 Hydrological reconstruction using the rainfall-runoff modelling approach and comparison with the varved-based reconstruction

Naskaupi River Q-mean and Q-max (Fig. 8) were also reconstructed using the ANATEM rainfall-runoff modelling (Fig. 10). The independent modelling approach results show similarities with reconstructions based on varved series. The ANATEM reconstructions are statistically and positively correlated with the yearly time series obtained from combined DLT and $P99D_0$ series during the 1880-2011 period (Q-mean: r = 0.41; Q-max: r = 0.22; n = 131; p < 0.01). The reconstructed Q-mean and Q-max annual variabilities show similarities, especially during the 1973–2011 period (Q-mean: r = 0.58; Q-max: r = 0.34; n = 43 p < 0.05).

Q-mean reconstructions with both varve parameters and modelling are better correlated than the Q-max reconstructions. This may be due to the higher uncertainty related to the Q-max reconstruction with the modelling approach. Indeed, high flow modelling requires good reconstruction performances on several hydro-climatic processes (i.e., snow accumulation during the winter, timing of the snowmelt, spring precipitation). Moreover, the uncertainty of the hydrological reconstruction is less important on recent periods (>1950), due to the better quality of the geopotential height field reanalysis over recent decades, as more stations series are available and thus used in the reanalysis. The decrease

in the uncertainty related to reanalysis over time might explain the better correlation
between the two approaches for the recent period.

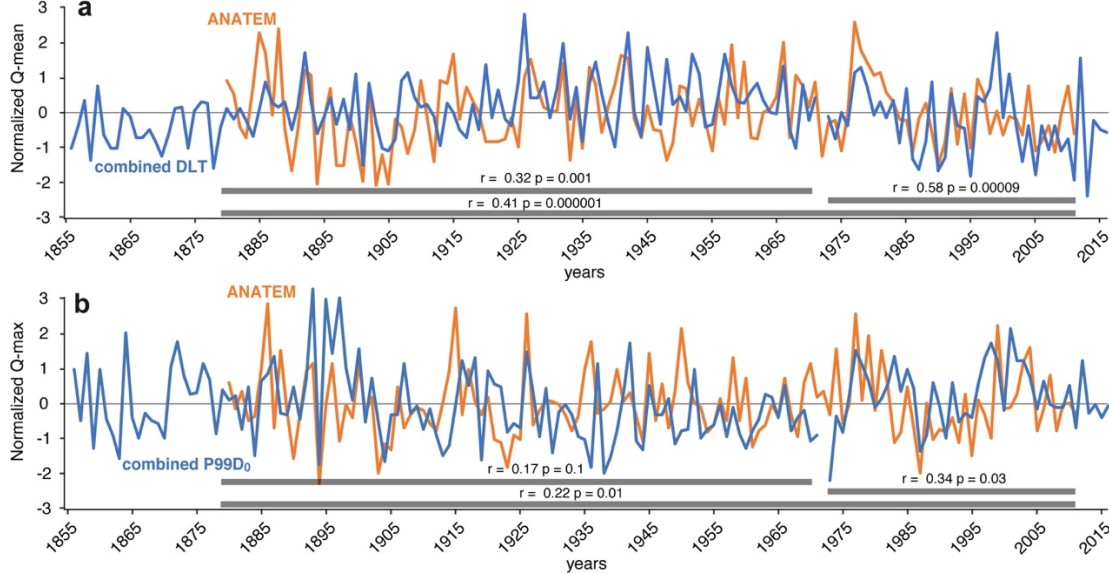


*Figure 10. Comparison between the Naskaupi River (a) Q-mean and (b) Q-max reconstruction using*
*combined Detrital Layer Thickness (DLT) (without the NAS-1 1972-2016 period) and P99D$_0$ series*
*respectively (blue line) and the rainfall-runoff modelling (orange line) for raw yearly data.*
**5. Discussion**
**5.1 Grand Lake varve formation**
Lakes containing well-defined and continuous varved sequences that allow the
establishment of an internal chronology are rare in boreal regions. However, the great depth
of Grand Lake, the availability of fine sediments in its watershed due to the glacial and
postglacial history of the region (Trottier et al., 2020), as well as its important seasonal
river inflow have favoured the formation and preservation of exquisite and thick varves.
The seasonal streamflow regime plays a significant role in the annual cycle of
sedimentation in Grand Lake and is responsible for the formation of the three distinct varve
layers. Due to the thickness and the clarity of the varve structures, it is possible to infer the
deposition mechanism for each layer and the season in which they were deposited.

The early spring layers are interpreted to be deposited during the river and lake ice break-
up and disintegration period, when erosion and resuspension of fine-grained sediments are
initiated but still low. Available Landsat-8 images of Grand Lake covering the 1983-2018
period (courtesy of the U.S. Geological Survey) shows that Grand Lake ice cover starts to
melt at the Naskaupi and Beaver River mouths. This ice melting pattern creates open bays
where drifting floating ice melts, thus depositing ice-rafted debris (Lamoureux 1999, 2004)
as observed in the early spring layer facies. The overlying detrital layers are interpreted as
flood-induced turbidites deposited at the lake bottom during the open-water season. High
energy sediment-laden river flows produce hyperpycnal flows allowing silt and sand-size
sediments to reach the cored sites (Cockburn and Lamoureux, 2008). The sharp contact
boundary between the early spring layer and the detrital layer at the top part of the early
spring layer supports the hypothesis that the detrital layers originate from underflows
(Mangili et al., 2005). The sediment waves on the Naskaupi and Beaver river delta slopes
(Trottier et al., 2020) (Fig. 1b, c) also indicate significant downstream sediment transport
by supercritical density flows (Normandeau et al., 2016). The thick and grading upward
basal part of the detrital layers are deposited during the high spring discharge period
generated by snowmelt runoffs. The lack of erosion marks between the early spring layer
and the detrital layer and the incorporation of rare cohesive sediment clasts within the
detrital layer suggests that erosion of the underlying early spring layers occurs in more
proximal and energetic settings. Three observations justify the combination of varve
measurements from the 3 coring sites : 1) the sedimentary processes inferred from the
observation of thin-sections, the high resolution bathymetric and the sub-bottom surveys
are similar; (2) the similarity of the varve facies and properties for each single year at the
3 different sites suggest a sedimentary pattern devoid of disturbances due to local factors;
(3) Grains-size differences are too subtle to infer different sedimentary processes and
environments. The upper part of varve structure in core NAS-1 show the most perceptible
different after 1972 (see discussion below). In spring, river discharge reaches its annual
peaks and sediment transport capacities that are then no longer reached during the rest of
the summer and autumn (Fig. 2, 3c, 11). However, the presence of thin coarser intercalated
sub-layers in the upper part of the detrital layer indicates that some rainfall events, as
observed in Fig. 11 (i.e., 1983, 1987, 1992, 1999) also contribute to deposition of sediments
in this layer. The overlying autumn and winter layer resulted from the settling and
flocculation of fine particles in non-turbulent condition from fall through the onset of lake
ice, forming a typical clay cap.

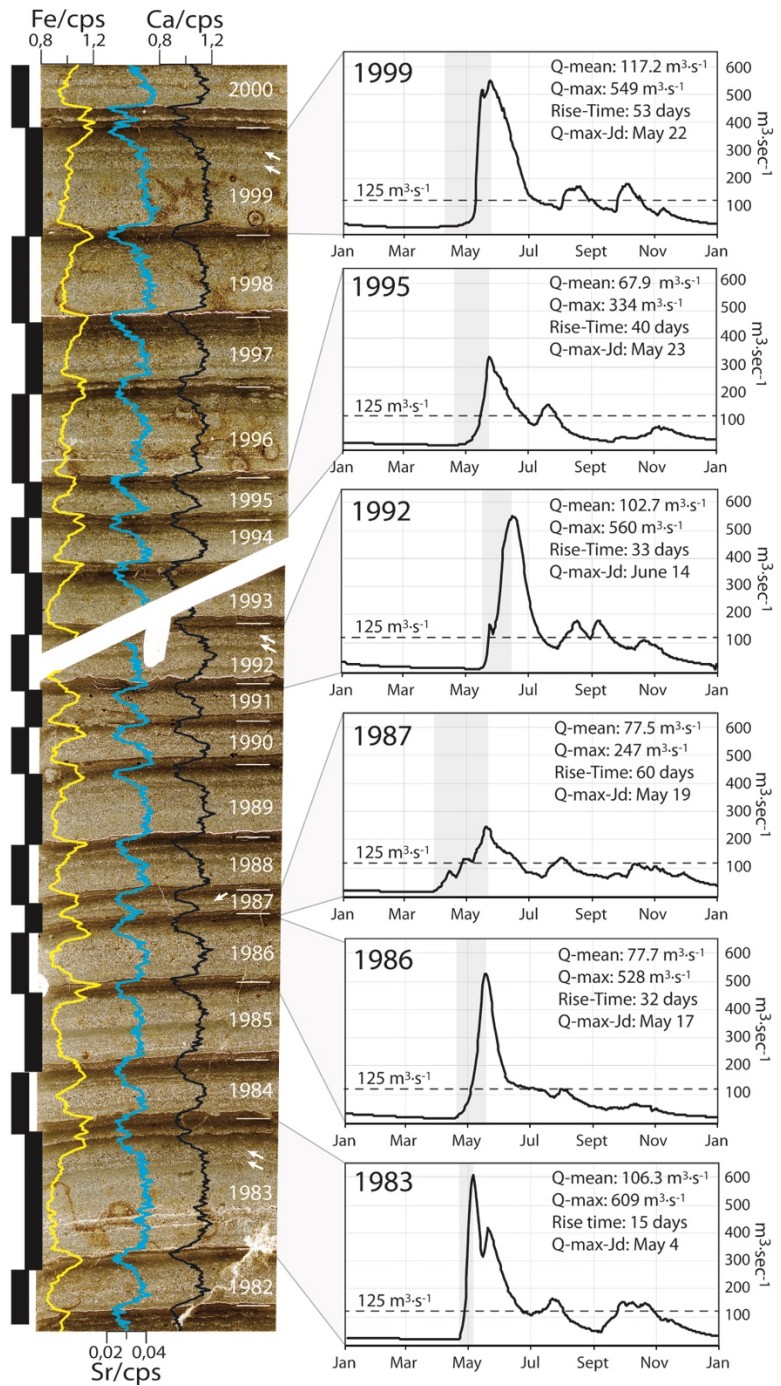


*Figure 11. Qualitative comparison between NAS-1A varves from thin-sections (delimited by the black bars)*
*with the hydrographs of the Naskaupi River. Observed annual Q-mean and Q-max as well as the timing and*
*rise time of the peak spring discharge are shown. Black dotted lines represent the discharge threshold of*
*~125 $m^3 \cdot sec^{-1}$. (1999, 1992, 1986, 1983) Strong spring floods associated with thick coarse varves. (1995,*
*1987) Low spring floods associated with thin varves. (1999, 1992, 1987, 1983) Coarser intercalated sub-*
*layers in the upper part of the detrital layer linked with summer and autumn high-discharge events. (1986)*
*Strong spring flood with a low summer and autumn flow associated to a varve without substructure. Thin-*
*sections are overlain by iron (Fe: yellow line), strontium (Sr: blue line), and calcium (Ca: black line) relative*
*intensities. See Fig. 5 for thin-sections locations.*

## 5.2 Anthropogenic influences on recent sedimentation

Anthropogenic environmental impacts on watersheds can be preserved in varved lake sediments (Zolitschka et al., 2015; Saarni et al., 2016; Czymzik et al., 2018). Changes observed in physical parameters of the varves deposited pre- and post-1971 at the NAS sites suggest that the effect of the dyke system on the Naskaupi River sediment inputs is perceptible in the Grand Lake varved sequence. The well-developed layers of varves deposited prior to 1972 from sites NAS-1 (Fig. 6b) and NAS-2, and the similarity between TVT and DLT values and variations among all sites over the 1856-1971 period (Fig. 6d) indicate that before the Naskaupi River diversion, seasonal sedimentation cycles appeared to have reached a relative state of equilibrium. The reduction of nearly half of the area of the Naskaupi River watershed due to its diversion in April 1971, reduced the water inflows and changed the base level of the downstream river system. The rapid base level fall must have triggered modifications of the fluvial dynamics from late-spring to winter 1971 (i.e., channel incision, bank destabilization, and upstream knickpoint migration), likely increasing the availability of sediments in the river system. The Naskaupi River spring/summer/autumn flood(s) of 1972 have then remobilized and transported a large amount of newly available floodplain sediments. This major sediment discharge plunged in Grand Lake and extended as hyperpycnal flow in the axis of the Naskapi River depositing a thick and coarse-grained turbidite following the lake bathymetry. This 1972 marker bed suggests that the Naskaupi River diversion had an impact on sedimentation at sites NAS-1 and NAS-2.

The increase in thickness and particle size values of varves deposited post-1971 in core NAS-1 (Fig. 5a, 6d/e, 7b, 11) suggest that the diversion has affected sedimentation at this site over time. During the 1972-2016 period, the river floodplain morphology must have been in a re-equilibration phase favourable to erosion, sediment transport, and deposition of coarser varves on the Naskaupi River delta slope. Since the river diversion, sedimentation at NAS-1 site appears to have become more sensitive to maximum discharges variations in spring than mean annual discharges. The sensitivity of the more proximal NAS-1 site to Naskaupi River extreme discharges variability may partly explain why better results are obtained without the 1972-2016 period to reconstruct Q-mean and

by keeping this period to the Q-max reconstruction. The negative correlation between
$P99D_0$ of the core NAS-1 and the timing and rise time of spring discharge (Table 3) also
demonstrate reactivity to spring entrainment energy conditions at this site. The distal NAS-
2 site shows that post-1971, sedimentation seems to have slightly lost sensitivity to river
discharge, and that sediment input continued to decline at the beginning of the deep lake
basin. The thin early spring layers free of ice-rafted debris in varve post-1971 of core NAS-
1 (Fig. 5a, 11) and NAS-2 indicate that the capacity of early spring discharge to transport
fine sediments and its ability to float ice to Grand Lake decreases along with the decrease
in water supplies.

It is tempting to link the decrease of varve thickness in core NAS-2 over the 1972- 2016
period with the discharge reduction due to the river diversion. However, similarities with
core BEA-1, a site devoid of anthropogenic perturbations (unaffected by the Naskaupi
River diversion) which also shows a decline in varve thickness, suggest that this decrease
can potentially be due to natural hydro-climatic conditions. The observed Naskaupi River
Q-mean series also show a decrease on the 1978-2011 period. Indeed, because of the distant
location of site BEA-1 from the Nakaupi River mouth, the diversion is most likely not
responsible for the decrease of varve thickness in this sector. Moreover, it is quite unlikely
that the sedimentary input from the Naskaupi River contributed to sediment accumulation
at the mouth of the Beaver River. The absence of any traces of the 1972 marker bed at the
Beaver River mouth (BEA-1) supports this hypothesis. Furthermore, the thickness decrease
observed in BEA-1 began after ~1920 (Fig. 6a), which is before the 1971 diversion.

Anthropogenic modification of the Naskaupi River watershed makes it challenging to
discuss natural hydroclimate-related variations before and after 1971. Some caution should
be applied when comparing pre- to post 1972 reconstructions, given the changes in
watershed conditions that happened after the construction of the system of dykes. There is
no instrumental data available for the Naskaupi River watershed before 1971 to confirm
that the calibration model post-diversion (1978-2011) is similarly robust for the preceding
period. The river diversion affected the Naskaupi River sedimentation dynamics but did
not modify it drastically. Despite the observed post-diversion changes in varves' physical
parameters in cores NAS-1 and NAS-2, which are however moderate, the varves still
responded directly to variations in river discharge. In addition, the part of the watershed
that has been diverted is an area composed mainly of lakes, which are not very
hydrologically reactive.
**5.3 The hydro-climatic signal in the varve record**
The significant correlations between continuous varve thickness and particle size
measurements with instrumental hydrological variables (Tab. 3) show that Grand Lake
varved sediments are reliable proxies to reconstruct past hydrologic conditions through
time at the annual to seasonal scale. The thick and/or coarse-grained varves correspond
well to years of high river discharges, whereas thin and/or fine-grained varves are related
with years of low discharge. Moreover, figure 11 clearly demonstrates how Grand Lake
varve record can be exploited to examine the interaction between meteorological
conditions and rivers discharge at an inter-seasonal scale, which is a temporal resolution
rarely obtained with natural proxies.

Data from the 3 sites were combined in order to better capture the regional hydroclimatic
signal and to somehow attenuate the noise that is inherent from the analysis of a single core
in a very large lake. A single core will be more sensitive to local specificities and is
probably less representative of the entire hydrogram. The Beaver and the Naskaupi Rivers
have adjacent catchments that share the same climatological and geological characteristics,
while the Beaver River's catchment is devoid of anthropogenic modifications. The
combination of varve parameters from different coring sites with distinct sediment sources
(Fig. 1b) improved the correlations with local and regional hydrological variables (Tab. 3)
and thereby the reconstructions (Fig. 8, 9). By integrating the core BEA into the combined
data, it allows to capture the hydrological signal from a larger region (Nakaupi + Beaver
watersheds) and it helps to capture the natural hydrological signal in our combined series
used for reconstructions.

As demonstrated by previous studies on varved sediments, the use of both varve thickness
and particle size analysis allows for a more specific investigation of the range of
hydroclimate conditions recorded within varves (Francus et al., 2002; Cockburn and
Lamoureux, 2008; Lapointe et al., 2012). For Grand Lake, the combined DLT is found to
be the best proxy to reconstruct all hydrological events occurring throughout the year (Q-
mean). DLT series are better at predicting Q-mean because the early spring layers and
autumn and winter layers thickness are more variable and are included in the TVT
measurements. This variability can be linked to specific climatic and geomorphological
parameters such as the duration of ice cover on Grand Lake and the Naskaupi River ice
breakup processes which induce noise in the hydrologic signal contained in TVT series.
The combined $P99D_0$ yields the strongest correlation in our dataset (Tab. 3) and is the best
proxy to reconstruct maximum annual discharges (Q-max). This result is logical because
the peak discharge is controlling the competence of the river and consequently the size of
the particles that can be transported. Moreover, this indicator is not sensitive to sediment
compaction, which may affect other proxies based on thickness.

The significant positive correlations between varve physical parameters and Snow-Win,
Q-nival (Tab. 3) and even Temp-Spring demonstrate that Grand Lake varve predominantly
reflects spring discharge conditions (e.g., Ojala and Alenius 2005; Lamoureux et al., 2006;
Saarni et al., 2016; Czymzik et al., 2018), which is the major component of the regional
streamflow regimes classified as nival (snowmelt-dominated) (Bonsal et al., 2019). In
boreal regions, the intensity and length of spring floods are controlled by the snow
accumulation during winter and by the temperature of the melting period (Hardy et al.,
1996; Snowball et al., 1999; Cockburn and Lamoureux, 2008; Ojala et al., 2013; Saarni et
al., 2017). The negative correlation between $P99D_0$ of the NAS-1 and the timing and rise
time of spring discharge suggests that early spring flows that increase rapidly are conducive
conditions for high entrainment energy and the deposition of coarser laminations on the
distal part of the delta slope (Fig. 11; site NAS-1). The erosion of detrital materials in early
spring increases when the snowmelt runoffs occur on soils that are not yet stabilized and
protected by vegetation (Ojala and Alenius 2005, Czymzik et al., 2018).

Intercalated sub-layers in the upper part of the detrital layer are interpreted to be produced
by summer or fall rainfall events (Fig. 11). Yet, the significant positive correlations
between varve thickness and Nb-days-SupQ80 suggests that a daily discharge of ~125 $m^3 \cdot s^{-1}$
represents an approximate threshold above which the deposition of coarse sediments in
Grand Lake (detrital layers) is more likely to occur (Fig. 11) (e.g., Czymzik et al., 2010,
Kämpf et la., 2014). According to the instrumental data (Fig. 2, 11), such a discharge can
be generated during the summer/autumn period, confirming that rainfall events can indeed
be triggering the deposition of thin intercalated sub-layers observed in the upper part of the
detrital layers (Fig. 11). However, there is non-significant low correlations between varves
thickness and Ptot-Annual/Ptot-Sum (not shown) suggesting that rainfalls contributions to
TVT remain small. These rainfall events have no contribution to $P99D_0$ because the
coarsest particles are found at the base of the detrital layers.

The comparison between the Naskaupi River hydro-climatic variables and other Labrador
hydrometric stations (Fig. 3) show that a coherent regional hydrological pattern exists in
the Labrador region. The performed regional Q-mean reconstitution and validation (Fig. 9)
indicated that the Labrador region hydrologic signal is recorded in the Grand Lake varve
sequence. The local and regional Q-mean reconstructed from the combined DLT series
(without the NAS-1 1972-2016 period) suggest a statistically significant decreasing trend
in mean annual discharge during the last 90 years. Naskaupi River Q-mean and Q-max
reconstructions based on both varve series and rainfall-runoff modelling revealed high
value periods from 1975 to 1985 and 1995 to 2005, and low values from 1986 to 1994 and
2006 to 2016 (Fig. 10). These results agree with the downward trend of the annual
streamflow observed in eastern Canada during the 20[th] century in other studies and also
with the reported higher river discharges from 1970 to 1979 and 1990 to 2007, and lower
discharges from 1980 to 1989 (Zhang et al. 2001; Sveinsson et al., 2008; Jandhyala et al.,
2009; Déry et al., 2009; Mortsch et al., 2015; Dinis et al., 2019).

In addition to providing a new high-quality varved record in eastern Canada, this research
highlights the complementarity between palaeohydrological reconstructions extracted
from clastic varved sediments and rainfall-runoff modelling. Both methods independently
offer a similar, yet robust, centennial perspective on river discharge variability in an
important region for the economic and sustainable development of water resources in
Canada. Reconstructed long-term mean and maximum annual river discharges series
provide valuable quantitative information particularly for water supply management for
hydropower generation and the estimation of flood and drought hazards. The varved
sediment of Grand Lake also allows documenting the effect of dyke systems on the
downstream sediment transport dynamic into a watershed and its implication for
palaeohydrological reconstruction. Further investigation of the impacts of the Naskaupi
watershed reduction on sediment transport could help better refine these reconstructions.
Future work in Grand Lake should be directed towards the high-resolution analysis of long
sediment cores in order to produce longer reconstructions. The Grand Lake deeper varved
sequence potentially recorded the hydro-climatic variability that occurred during the Late
Holocene in region sensitive to the North Atlantic climate, allowing interesting prospects
into large-scale atmospheric and oceanic modes of variability.

**6. Conclusions**
The great depth of Grand Lake, the availability of fine sediments along its tributaries, and
its important seasonal river inflow have favoured the formation and preservation of fluvial
clastic laminated sediments. By using a new varved record in eastern Canada and a rainfall-
runoff modelling approach, this paper provides a better understanding of the recording of
hydro-climatic conditions in large and deep boreal lakes and allows extending the
hydrological series beyond the instrumental period as well as the spatial coverage of the
rare annual palaeohydrological proxies in North America. The key results of this study are:
• The annual character of the 160 years-long lamination sequence has been confirmed.
Each varve, composed of an early spring layer, a summer/autumn detrital layer and an
autumn and winter layer, represents one hydrological year.
• Grand Lake varve formation is mainly related to the largest hydrological event of the
year, the spring discharge, with minor contributions from summer and autumn rainfall
events.
• Two hydrological parameters, the Naskaupi river Q-mean and Q-max annual
discharges, are robustly reconstructed from two independent varves physical
parameters, i.e., the detrital layer thickness (DLT) and grain size ($P99D_0$) respectively,
over the 1856-2016 period. The reconstructed Q-mean series suggest that high Q-mean
years occurred during the 1920-1960 period and a decrease in Q-mean takes place
during the second half of the 20[th] century.
• The same two hydrological parameters (Q-mean and Q-max), were also reconstructed
using the ANATEM rainfall-runoff modelling. ANATEM discharges series show
similarities with reconstructions based on the varved series, which support the
reliability of the two independent reconstruction approaches.
• The statistically significant relation between combined DLT series and the observed
Labrador region Q-mean series, extracted from five watersheds of different size and
location, demonstrates that Grand Lake varved sequence can also be used as a proxy of
regional river discharges conditions.
• The effects of Naskaupi River dyking in 1971 are clearly visible in the sedimentary
record and affected sedimentary patterns afterwards. While this event makes the
hydroclimatic reconstruction trickier, it remains that the outstanding quality of this
varved sequence provides one of the best hydroclimatic reconstruction from a
sedimentary record, with Pearson correlation coefficients up to $r = 0.75$.

## Data availability

The data set used in this study will be available on the PANGAEA database.

## Author contributions

This study is part of AGP's thesis under the supervision of PF and PL. AT and PL provided geophysical data (Fig. 1b, c) and useful information on the morpho-stratigraphical framework of Grand Lake. AGP and DF conducted the coring fieldtrip. AGP and PB collected instrumental data. PB calculated hydro-climatic variables from instrumental data (Fig. 3) and performed the rainfall-runoff modelling. HD and AGP adapted the code used to establish the relationship between the varve parameters and the instrumental data and for the regression model. AGP performed most of the data analysis, wrote the manuscript and created the figures with contributions from PF and PB. All authors provided valuable feedback and contributed to the improvement of the manuscript.

## Competing interests

The author Pierre Francus is a member of the editorial board of the journal.

## Acknowledgments

This research was financially supported by NSERC-Ouranos-Hydro-Québec-Hydro-Manitoba through a CRD grant to P.F. and P.L. (PERSISTANCE project, É. Boucher et al.). This work was also supported by the FRQNT through a doctoral (B2X) research scholarship to A.G.P. and by the MOPGA Short Stay program grant at Université Côte d'Azur, Nice, France to A.G.P and P.B. A financial support for the fieldwork campaign at Grand Lake was provided by POLAR through the NSTP program to A.G.P. The authors are grateful to Arnaud De Coninck, David Deligny and Louis-Frédéric Daigle for their participation during fieldwork, laboratory and helpful discussions. We greatly thank Wanda and Dave Blake from North West River for their guiding experience and accommodation at Grand Lake. We thank the Labrador Institute at North West River for the use of their facility during fieldwork. We want to thank Stéphane Ferré from the Micro-Geoarchaeology Laboratory of the Center for Northern Studies (CEN) in Québec, QC, Canada, for the production of the high-quality thin-sections used in this study. We would

also like to thank the three reviewers for their constructive review of this article. Finally,
many thanks to Monique Gagnon, Charles Smith and Clarence Gagnon for reviewing the
English of an earlier version of the manuscript.

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
