# Peer review of "Reconstructing past hydrology of eastern Canadian boreal catchments using clastic"

_Climate of the Past, 2020_

## Referee Comment (RC1) · Krystopher Chutko (Referee) · 17 Aug 2020

This paper presents a short varve chronology from Labrador along with various hydro-climatic interpretations. The identification of varves in this particular region is important due to the limited availability/identification of palaeoenvironmental proxies in the Boreal region of eastern Canada. This work proposes to help fill that gap. The authors suggest a potential for a longer-term record to emerge from this lake – this would greatly benefit hydro-climate reconstructions in this region. The palaeohydrologic interpretation of the varve record is robust, supported by independent dating and multiple statistical

approaches.

Overall, the sedimentary analyses and interpretations are sound. Most of my comments below focus on the reporting of the statistical analyses. The figures are well-drawn. There is a heavy reliance on acronyms which take some time to get familiar with. In many places, a comma is used instead of a period for quantities (eg, Fig 9 vs Table 1) – from a Canadian perspective it doesn't matter which is used but pick one convention for consistency. Four research objectives are identified in the introduction, and the paper discusses each of these sufficiently.

I would recommend publication of this manuscript with the below comments/suggestions/questions addressed.

Specific comments:

Line 111-113: how are "winters" and "summers" defined? Later in the paragraph the snowmelt season is defined as AMJ, but there is no similar definition of the seasons. Assume JFM and JAS?

Line 189: "counts were executed repeatedly". How were the counts made? Multiple counters? Multiple counts per counter? There is a mention of counting difficulty (line 382). If multiple counts were made, how consistent were those counts? Given the clear images and laminae it would seem to be fairly clear-cut, but I'd like to see some mention of the accuracy/precision of the counting process to fortify that.

Line 244-245: Only 1 of the 5 instrumental records goes back to 1966 (incomplete data 1966-68?). Is this good enough to extend the composite instrumental record back to 1969? "Strong positive correlations" are stated but not shown – could these be added to Table 2? Also, the extension crosses pre- and post-diversion boundary – is it still reasonable extend the record back past 1971?

Line 252: linear regression models. "simple linear regression" is used to model the relationship between varve thickness and hydrometric variables. Adjusted R2 is listed

as an evaluative statistic. Adjusted R2 should be reserved for multiple regression, since it adjusts the coefficient based on the number of independent variables. With only one independent variable, the unadjusted R2 is appropriate (listed as Multiple R-squared in R). Similar with Figs 8 & 9.

Line 371-374: This triggered a flag for me – why did the 1971 changes result in a thick and coarse unit? It is explained later on (section 5.2) but left me wanting more explanation here in the results section.

Line 411-: a lot of p-values shown here using a 0.05 threshold (and Table 1). This defeats the purpose of using p-values which are intended to show the actual probability of attaining the particular statistic. Really this is just the same as accept or reject at 95% confidence, which is far too arbitrary. Can these threshold values be replaced with actual p-values to make the analysis more objective? To make matters worse, the threshold value changes to 0.01 in Fig 6. Reporting actual p-values will help with consistency. In line 435-438 there are several r values with no p-value attached. They are "significant" correlations, but no indication of how significant. I would suggest actual p-values to 3 decimal places would suffice.

Line 474: "1972 is considered as an outlier". Is this a subjective consideration or is it supported by the statistical analyses? For example, does the leverage for 1972 appear high when evaluating the regression analyses?

Technical corrections: remove/add what is in [ ]

Line 32: take[s]

Line 69: method[s]

Line 79: switch "into" and "the" around

Line 135: [a]eolian [this is very picky]

Line 157: [an] undisturbed or undisturbed area[s]

Line 211: Using [a] custom

Line 227: replace indice with index

Line 244: allows [an extension to the] instrumental

Line 249 Table 2: km2 - add superscript

Line 255: Model[s]

Line 275: station[s]

Line 279: "thanks to the. . ." – this is rather informal compared to the rest of the writing. Change to "using the Oudin et al. . ."? Same on line 304.

Line 378-379: structures allowed [to build] a robust age-model reproducible among cores [to be constructed].

Line 379: why is the $1 - 5$ km distance "significant"? Significant with respect to what? Suggest removing the word.

Line 392: ([F]ig. 6a)

Line 401/415: "slight" – what does this mean? Can this decrease in TVT/DLT be supported statistically?

Line 444: [since]

Line 490: 1887-1991 – should this be 1887-1891?

Line 491-493: this sentence is incomplete. Perhaps solved by removing the "While" at the beginning.

Line 500: varve[s]

Line 514: replace on with for

Line 538/589: important. What does this mean? It seems to be used as a synonym for

significant, but it doesn't fit well. The sentences work without the adjective.

Line 552: Beaver[s]

Line 583: "floods of [the years] 1972 CE [has (have)] remobilized"

Line 588: bank[s]

Line 589: [r]iver

Line 595: replace for with to

Line 625: good – another of those pesky vaguely meaningful words. What does it mean in this case – what is a good correlation? Can 'significant' be used here instead?

Line 634: global. Do these cores contain a global hydro-climatic signal? Or is it regional (see line 92)?

Line 685: recorded in [the] Grand Lake. . .

Line 699: discharge[s]

Line 746: record[s]

---

## Referee Comment (RC2) · Anonymous Referee #2 · 22 Aug 2020

General comments:

The manuscript by Gagnon-Poire and co-authors entitled 'Reconstructing past hydrology of eastern Canadian boreal catchments using clastic varved sediments and hydroclimatic modeling: 160 years of fluvial inflows' presents river discharge reconstructions from three short cores containing clastic varves reaching 160 years back in time. For the discharge reconstruction mainly two proxies have been applied (grain size and layer thickness). These data demonstrate the large potential for discharge reconstructions using annually laminated sediments. However, a few week points in the interpretation

need to be better clarified. In general, it is difficult to follow the large number of different statistical correlations between cores, proxies, proxy reconstruction and model results. A more concise approach with a focus on main correlations would make the manuscript easier to read. Furthermore, instead of levelling out the different signals in the three cores by a pooling approach, the causes for these differences should be better examined and documented. The implications of the difference between cores for selecting the most suitable core location for palaeoydrological reconstruction should be elaborated.

The cores have been taken from different parts of the delta surface and even the most distal core location is still 70 m above the deep basin. Sediment reworking processes on the delta should have an influence on the deposition and layer thickness as well as grain size. For example, a thinning of discharge layers from the proximal to the distal delta location (NAS-1 to NAS-2) should be expected, which, however, is not seen in the layer thickness plots shown in figure 6. A more detailed discussion of sedimentological processes on the delta surface should be added for clarification.

The 'anthropogenic impact' after dyke construction (in 1971 or 1972?) has been stressed several times (e.g. lines 444/445). However, it is not clear how exactly dyke construction impacted on the sedimentation. Was the main effect generated by the earth movements during dyke construction (if at all, how long did his effect last?) or by the reduction of the catchment? If dyke construction resulted in 'increased availability of sediments in the river system' as suggested (lines 588-589), why is that only seen in NAS-1 core? Why should there be more sediments in the system although the catchment size decreased? The different behavior of the cores NAS-1 and NAS-2 after 1972 need to be better elaborated. The argumentation that NAS-2 behaves like BEA-1 (lines 598 and following) is not convincing because the BEA-1 location is not affected by the Naskaupi River inflow, whereas NAS-1 and NAS-2 are located in the same direction towards the river inflow. Furthermore, in contrast to DLT, grain-size data do not show major difference between both cores after 1972. How is that explained?

Due to the core differences, post 1972 DLT data of NAS-1 were excluded from statistical analyses? Instead of excluding the data, correlation of NAS-1 and NAS-2 core data post 1972 with hydrological data should be compared. It would be interesting to see how the sedimentological differences affected the correlations with hydrological data.

The proxy data from different cores have been pooled to obtain a better statistical correlation with hydrological variables (lines 630-631). However, pooling masks the different sensitivity of the different core locations in recording natural hydrologicial variability. Moreover, it is not clear if the pooling includes all data from all cores or if some parts of the data are excluded. In line 614 it was pointed out that the post 1972 period has been excluded from one of the cores (NAS-1). If this part of the record is also not included in the pooling approach you put apple and pears in the basket and I wonder about the meaning of improved statistical correlation. Since the BEA-1 and NAS-1 (lines 599-604) are considered to record the 'natural hydro-climatic signal' one should expect a better representation of palaeohydrolgical changes in one of these cores rather than in pooled data from all cores.

The authors report variability on different time scales, i.e. long-term trends in mean annual discharge (line 687) and decadal-scale variability (e.g. lines 56-57) but they do not explicitly relate these. The appearance of variability at different time scales is an interesting finding that should be more emphasized and elaborated in the paper.

The statement about dyke effects on sediment transport and its 'implications for palaeo-hydrological reconstruction' (lines 703-705) and that dyking effects are 'clearly visible in the sedimentary record' (lines 743-744) are too much simplified. It has been shown that one coring sites has been affected by dyke construction but the two others not or only to a minor degree. This differentiation between core locations is an important point and knowledge about these differences and their causes is essential to select the most suitable coring locations for palaeohydrological reconstruction. In this respect, and here I repeat my previous comment, I do not consider the pooling as suitable approach even if it may improve statistical correlation. Often unspecific terminology is

used like, for example, 'thick and coarse', 'thicker' (examples in specific comments). This should be changed into quantified information.

Specific comments:

A number of 'distinctive marker layers' (labelled A-P, Figure 4, lines 381, 382) have been defined but it is not explained how distinctive these layers are and what makes them distinctive. In figure 4 they do not appear distinctly different neither in the core image nor in the XRF data.

In the chapter 'Regional setting' some information about vegetation cover should be added since that may influence catchment erosion and clastic sediment transport into the lake.

In chapter 4.7 it is not clear which sediment proxies have been compared with the rainfall-runoff modeling approach. Are these proxy data from individual cores (which?) or from pooled data? If it is pooled data, how did you account for differences in TVT between cores?

Line 162: It should be specified which efforts were made to retrieve undisturbed sediment surfaces. Taking short cores from such deep lakes without disturbance is a common problem to the community and it would be helpful to know how the authors tried to improve the coring in this respect.

Lines 185-186: Sampling intervals for Cs-dating are unclear. Was it attempted to sample individual varves or only sublayers? Sample intervals vary between 2 and 0.5 cm but according to figure 6 layer thickness was > 4cm? Please clarify.

Line 226: Specify 'coarse debris' and quantify grain sizes

Line 227: Explain the PSI. Is this a mean grain size for each lamination? What is 'lamination' in this respect? A varve or a sublayer (which?)?

Line 325: What is 'occasionally'? Provide the number or percentage of DL with sharp

lower boundary.

Line 327: Explain 'non-annual' for these layers. All three described sub-layers (ESL, DL, AWL) are seasonal, i.e. non-annual. Also quantify 'thin coarser'. What is the thickness (range or mean) and grain size of these layers? Finally, quantify 'some cases', i.e. how many of these layers did you count?

Lines 328-329: Provide information why Ca and Sr are relatively higher in DLs, i.e. which minerals in the DLs include these elements?

Line 344: 'thick and coarse' is unspecific. Provide information about thickness and grain size of this prominent layer. Are there distinct differences also in the elemental composition of this layer?

Lines 349/350/351: the ESL of pre-1972 CE is 'thicker'. Provide quatified information instead of this unspecific information. It should be easy to calculate mean contribution of the ESL (in %) to the total varve thickness for the pre- and post-1972 intervals

Lines 350, 352: 'post-1971' or 'post-1972'?

Lines 372/373: When exactly was the anthropogenic change in the catchment? Was it in the year before the 1972 marker layer or in 1972? If it was in the year before, why was there a 1 years delay in the sediment response?

Figure 6. Add the position of marker layers A-P in the figure.

Lines 414 and following: How is the P99D0 value influenced by the ratio DL/TVT?

Line 550: How often is 'seldom'? In how many layers erosion traces have been observed.

Line 550/551: What kind of traces of erosion are these. Provide a description. I would expect differences between the proximal and distal cores. Please clarify.

Line 580: I disagree that river sediment input was 'quantitatively and spatially constant' before 1971. There is distinct variability at different time scales in the data, e.g. between 1920 and 1960s.

Line 602-604: It is assumed that 'natural hydro-climatic signal' drives the sedimentation in BEA-1 (and NAS-2) without saying what this 'natural hydro-climatic signal' is. This statement should be easy to be proven or disproven by correlation with instrumental hydrological data.

Line 634: You will get at best a regional hydro-climatic signal but certainly no global.

Line 642: Quantify 'slight variability'

Line 648: How do you explain 'high thickness values' (need to be quantified!) of ESLs and AWLs during the 1920s?

Lines 675-677: There is a detailed discussion on thresholds and flood amplitude re-construction in Kaempf et al., 2014 (J. Quat. Sci.) that you may consider including in this part of the discussion.

Technical corrections:

Lines 328-329: 'abundance of elements'. This is wrong because XRF scanner data are relative variations of element intensities but not quantified amounts

Line 547: instead of 'underlying' it should be 'overlying'

Line 571 (figure caption): see comment above, XRF data does not give 'abundances'. This are relative changes of element intensities

―――――――――――――――――――――――

---

## Referee Comment (RC3) · Anonymous Referee #3 · 26 Aug 2020

This study by Antoine Gagnon-Poiré and colleagues entitled "Reconstructing past hydrology of eastern Canadian boreal catchments using clastic varved sediments and hydro-climatic modeling: 160 years of fluvial inflows" presents an interesting counterpart to rainfall-runoff modeling approaches that aim at expanding instrumental streamflow datasets for multi-decadal analysis of hydrological variability. Indeed, this study based on varved sediment sequences aims at producing long river discharge records (>100 years) to support, help refine or contradict paleo-hydrological records offered by the modeling approaches.

[Figure]

The strength of this study is clearly provided by the very high-quality analysis of the varve record and the robustness of the sediment chronology. Varve boundaries are clearly defined through high-quality startigraphical analysis combined with CT images and state-of-the-art microscopy-based grain size analysis. Varve counts are consistent between the cores of different locations, and they are supported by independent 137Cs dating. The varve record thus offers an annual view into past changes without chronological constraints, which is a major advantage for developing a proxy-climate or proxy-hydrology models.

Varve stratigrahical analysis further allowed to select the best varve parameter (i.e., meaningful season) to compare with hydrological data. The proxy-hydrology correlations have been significantly improved by selecting the thickness of the detrital layer (DLT) instead of total varve thickness (TVT), thus reducing potential noise; spring discharge being the main driver for sediment erosion and transport in the nival catchment of Naskaupi River. In this context, Figure 11 is very stunning, and shows how a varve record can best be exploited to look at micro-meteorology and lower-than-seasonal resolution river hydrodynamics; this is novel.

However, although the quality of the sedimentary investigation is very robust, general important comments relate to the methods to produce the paleo-hydrological record and its regional signal. I hope that these major comments will be well received and accepted, and that they will be of good use to improve the present manuscript.

**General comments**

Normalizing total varve thickness (TVT) is interesting when several sediment cores are collected at the same location => thus to reduce local error in the proxy-hydro/climate relationship. However, merging TVT from a proximal (more sensitive, thus with larger amplitude) and distal record (buffering large changes in river discharge, recording annual change in hydrodynamics and only sensitive to the most intense discharge events) is neither properly justified in the text, nor fully appropriate. It gives the impression that

the different records were merged in the way that the correlation with hydrometric data would be maximize, at the cost of process understanding. A great example is losing the downward trend in TVT from NAS-2 by merging its record with NAS-1, which has no trend. The same applies to (and I would say particularly applies to) P99D0. Mean values are strongly driven by NAS-1, the proximal coring site. As such, it is not surprising to find the best correlation for Qmax to NAS-1 (proximal) and for Qmean to NAS-2 (distal). Overall, there is no mechanistic logical explanation in merging TVT, DLT or P99D0 from the three cores to help maximize the correlation. This is particularly the case integrating BEA core, for which it is argued (L604) that "it is quite unlikely that the sedimentary input from the Naskaupi River contributed to sediment accumulation at the mouth of the Beaver River" (i.e., BEA core). L443: There is no clear explanation on why the post-anthropogenic watershed modification would support the discarding of NAS-1 in the TVT, DLT and P99D0 normalization of the cores. It further supports the impression that the best records were merged in the way that the correlation with hydrometric data would be maximize, at the cost of process understanding. L461: Table 3 is named Table 1….it took me some time to realize that Table 3 was not missing, while being important and largely cited.

**General comment on the comparison between sedimentary data and hydrological variables**

Q vs SSC are always presented as a log-log linear regressions. The same should applied to DLT vs Q, likely to P99D0 vs Q. From the scatterplot presented in Fig 8, it is likely that the general proxy-hydrometric relation follows a DLT=f(log(Q)), or a log(DLT)=f(log(Q) relation rather than a linear relation. See Warrick (2015) and references therein, or Thurston et al. (2020). This should be tested as it has major implications on statistical yields in the sediment-hydrological relations.

Warrick, J. A. (2015). Trend analyses with river sediment rating curves. Hydrological Processes, 29, 936–949. https://doi.org/10.1002/hyp.10198

Thurston et al.    (2020).    Modelling suspended sediment discharge in a glaciated Arctic catchment–Lake Peters, Northeast Brooks Range, Alaska. https://doi.org/10.1002/hyp.13846

**General comment on the regionalization of the signal**

The merging of the different watersheds of the region is interesting, but I don't think that the quantitative analysis is relevant. This is exemplified by the low correlation of r=0.49 (even though significant) between the Naskaupi River and the Eagle station. This means that the discharge data from the Naskaupi River can only explain 24% of the variance in Eagle discharge data, independently from the sediment context. Removing Eagle from this merging exercise will not solve this issue. Each watershed is sensitive in its own way not only to specific climatic (evidence is missing that the climate in the Naskaupi region is representative of a broader region, not only through correlation between hydrometric station data) but also to geomorphic conditions that are not integrated into the daily climatic series of the CemaNeigeGR4J model (such as slope, erosion susceptibility, potential geological difference, orientation...), and that can differ significantly within the 500x500km grid used in this manuscript. A more detailed analysis of the different watershed, their runoff response (timing, strength, duration, sensitivity to snowmelt vs rainfall) would merit further investigation. L241: "These four streamflow series (Tab. 2) show strong positive correlations with Naskaupi River discharge", one expects to see these strong positive correlations. Figure 3 presenting the location of the different catchment for regionalization of the findings would have benefited an additional panel with daily streamflow time series for each catchment as in Figure 2, for instance.

**General comments on the calibration-in-time model**

A proxy-hydrology calibration model is built for the period 1978-2011, and reconstructed back to 1876. Post 1972 (River deviation) shows that the system has changed hydrologically with discharge reduced by a factor 2. This should also be true sedimentologically, and a few points are in line with this (contre-)hypothesis: clear change in the preservation of DLT in NAS-2, change in the mean P99D0 record of NAS-1, change in mean and variance of DLT and TVT of BEA-1, and most significant change for TVT and DLT post 1972 in NAS-2. These observations thus contradict the sentence L580 "River sediment input seems to have been quantitatively and spatially constant." The principle of stationarity being not respected hydrologically, it is doubtful that the calibration model post anthropogenic modification remains valid for the preceding period. Deeper discussion are required on this topic, e.g., by proposing evidence that the sediment record (through TVT, DLT, or best P99D0) is not significantly affected by this change and can be used to infer river hydrodynamics prior 1972.

L604: "it is quite unlikely that the sedimentary input from the Naskaupi River contributed to sediment accumulation at the mouth of the Beaver River" is in contradiction with L440 : "data from core BEA-1 (1856-440 2016), NAS-1 (1856-2016) and NAS-2 (1968-2016) have been normalized and averaged to produce mean TVT, DLT and P99D0 series" to be compared to the Naskaupi River hydrometric station. This questions the selection of BEA-1 in the merging approach of the sedimentary data.

Moreover, the justification that Naskaupi River discharge does not affect BEA-1 location is made by the fact that (L598-608) "the absence of any traces of the 1972 CE marker bed at the Beaver River mouth (BEA-1) supports this hypothesis." This argument is not admissible, especially with regards to the previous discussion (L583) that the "flood(s) of the years 1972 CE has (have) remobilized newly available sediments and deposited a thick and coarse-grained turbidite on the lake floor". It is indeed likely, with regards to the sedimentary facies of cores NAS, that the 1972 flood transported coarse material that plunged in the river proximal and extended as hyperpicnal flow following the lake Bathymetry (NAS-1 to NAS-2), thus not affecting BEA core location. However, discussion about flood hydrodynamics and annual river discharge in terms of sediment transport should be decoupled in the discussion.

The argument that a decline in varve thickness is also observed post 1972 in BEA, thus

related to a natural hydro-climatic signal can be true, but seems superimposed to the effect of the Naskaupi River diversion, especially for cores NAS. While discreet peaks of sediment proxy (TVT, DLT, P99D0) for the different sediment cores are consistent (occurring at the same date), the variance, mean, and trend in these data are not comparable enough to allow the merging. Also, the three records from the three cores respond totally differently to the pos-1972 hydrological changes: lower mean for BEA-1, higher mean and increase variance for NAS-1, lower mean + decreased variance + decreasing trend for NAS-2. Suggestion: change point analysis (mean, variance and trend) can be performed on each times series, both from the hydrological and sedimentary variables. This would give statistical support to visual information.

Finally, I am really surprised to see a 5-year running mean for the reconstruction of hydrological data. As the varve chronology is more than robust, through its coherence between the different locations and perfect correspondence with 137Cs, it is a pity that annual time series are not reconstructed. This choice of smoothing the data needs to be justified. Running mean in lake sediment studies are generally used to account for the error in the varve chronology, with statistical justification for significant improvement of the proxy-climate correlations (cf. Von Gunten et al., 2012). + Figure 10 compares the rainfall-runoff model and sedimentary data at annual resolution, with no lag (L624). This gives again the impression that correlation values are maximized at all cost.

von Gunten, L., Grosjean, M., Kamenik, C. et al. Calibrating biogeochemical and physical climate proxies from non-varved lake sediments with meteorological data: methods and case studies. J Paleolimnol 47, 583–600 (2012). https://doi.org/10.1007/s10933-012-9582-9

**General comment on the rainfall-runoff modeling approach**

A key point of this review is the comparison between sedimentary data and modeling. The rainfall-runoff modeling for each catchment is merge to a single ANATEM time series (Fig. 10) and compared to the sediment properties of the varves. This ANATEM

time series is based on the pre-determination of single catchment area, then extended for the whole studied period. However, the Naskaupi river watershed pre- and post-1972 is different (smaller after the 1972 river deviation) and should be adapted in the modeling; producing two time series (i) 1880-1971, (ii) 1973-2011. This likely explains that stronger correlations found between e.g., DLT and ANATEM for the period 1972-2011 (r=0.54) compared to the preceding period (r=0.31).

**Specific comments**

L68: to reconstruct daily. . .

L73: "Long hydro-climatic series based on natural proxies in the study region are rare and limited to tree-ring". What have all these studies produced? What conclusions ? Is the aim of the present study to comfort previous finding, to increase spatial coverage? This does not say why clastic lake sediment are better than tree rings or pollen data (which is suggested here) Aren't tree-ring records not enough? Are they all from the Labrador region? Are the hydroclimate records consistent with each others? Answering these question would help re-shaping the sentences in explaining what makes clastic varves so specific and powerful.

L76: "clastic" are not defined prior to this mention

L79: Remover 'The' between area and into

L81: Amann et la., should be et al.,

L231 : remove 'used' form the title

L245: Suggested change; 'This allows to extend instrumental data series for the period 1969 to 2011, and fill in data for the missing years.'

L252: title could be simply, e.g., varve properties and hydrological variables

L456: "data show significant (p < 0.01) strong positive correlation." Remove 'strong', especially referring to r = 0.49 in brackets, this is not a strong correlation, especially in

such hydrological context.

L478: " The significant correlation between reconstructed Q-mean and Q-max values and observed discharge data validates the predictive capacity of the model." I don't see how the fact that Qmean and Qmax correlates validates the proxy-Q model.

L496: "demonstrates that the Grand Lake varved sequence is robust and contains a regional signal." You mean the hydrological reconstruction is robust? I would not say that R2 = 0.41 is robust. Remove 'robust' and keep 'contains a regional signal.

Q-mean and Q-max are sometimes written with a capital M (e.g., Q-Mean), sometimes not (Q-mean). Please stay consistent.

L595: it should read 'indicate that the capacity of spring discharge to transport fine sediment and its ability to float ice to Grand Lake decreases due to the decrease in water supply.'

L650: please consider changing 'robust' for 'best proxy'

L697: extracted from

L706: change "could help to better our reconstructions" to 'could help better refine these reconstructions"

---

## Author Comment (AC1) · 2 Sep 2020

This paper presents a short varve chronology from Labrador along with various hydro-climatic interpretations. The identification of varves in this particular region is important due to the limited availability/identification of palaeoenvironmental proxies in the Boreal region of eastern Canada. This work proposes to help fill that gap. The authors suggest a potential for a longer-term record to emerge from this lake – this would greatly benefit hydro-climate reconstructions in this region. The palaeohydrologic interpretation of the varve record is robust, supported by independent dating and multiple statistical

approaches.

Overall, the sedimentary analyses and interpretations are sound. Most of my comments below focus on the reporting of the statistical analyses. The figures are well drawn. There is a heavy reliance on acronyms which take some time to get familiar with. In many places, a comma is used instead of a period for quantities (eg, Fig 9 vs Table 1) – from a Canadian perspective it doesn't matter which is used but pick one convention for consistency. Four research objectives are identified in the introduction, and the paper discusses each of these sufficiently.

I would recommend publication of this manuscript with the below comments/suggestions/questions addressed.

Reply: We thank reviewer #1 for his positive comments on our manuscript. The use of acronyms will be reduced to facilitate the reading of the manuscript. Also, a descriptive table grouping the main acronyms used in this study could be a good way to help readers familiarised with acronyms and make the reading more fluid. The used of the period for quantities will be uniformized.

Line 111-113: how are "winters" and "summers" defined? Later in the paragraph the snowmelt season is defined as AMJ, but there is no similar definition of the seasons. Assume JFM and JAS?

Reply: Winter (DJFM) and summer (JJA) will be defined in the revised version.

Line 189: "counts were executed repeatedly". How were the counts made? Multiple counters? Multiple counts per counter? There is a mention of counting difficulty (line 382). If multiple counts were made, how consistent were those counts? Given the clear images and laminae it would seem to be fairly clear-cut, but I'd like to see some mention of the accuracy/precision of the counting process to fortify that.

Reply: Due to the great quality of the varved sequences, two counts were made by one counter (AGP). As mentioned in the text, counting difficulties occur within varve years

1952-1953, 1935-1934, 1918-1919. The error percentage between the 2 counts will be mentioned in the text to further demonstrate the accuracy/precision of the counting process.

Line 244-245: Only 1 of the 5 instrumental records goes back to 1966 (incomplete data 1966-68?). Is this good enough to extend the composite instrumental record back to 1969? "Strong positive correlations" are stated but not shown – could these be added to Table 2? Also, the extension crosses pre- and post-diversion boundary – is it still reasonable extend the record back past 1971?

Reply: There is an error in the Tab. 2, the Eagle series goes from 1969 to 2016, not 1966 to 2016. This will be corrected. Dinis et al., 2019, produced an observed river index of summer regional discharge in Labrador for the 1969-2009 period. They normalised and average hydrometric data from the Eagle River; 1969–2009, Alexis River; 1978–2009 and Little Mecatina River; 1979–2009. As this paper has been accepted and published in the international journal Climate Dynamics, we think it is reasonable to use the same methodology to produce our Labrador region mean annual discharge series.

Dinis, L., Bégin, C., Savard, M. M., Marion, J., Brigode, P., and Alvarez, C.: Tree-ring stable isotopes for regional discharge reconstruction in eastern Labrador and teleconnection with the Arctic Oscillation, Clim. Dynam., 53, 3625-3640, https://doi.org/10.1007/s00382-019-04731-2, 2019.

The significant positive correlations between the four streamflow series (Tab. 2) with Naskaupi River discharge mentioned in the section '' 3.4 Hydro-climatic variables used" (line 242) are rather shown in the result section ''4.5 Relation between varve series and instrumental record" (line 455) because we believe that these are results. A note in section 3.4 will be added to guide readers (i.e. see section 4.5 for details on correlations).

We think that it is useful to extend the regional mean annual discharge series back past diversion boundary with the Eagle River hydrometric data because this produce

a longer calibration period using a large watershed of the Labrador region which is devoid of anthropogenic modifications. This also allows to calibrate the few varves pre-diversion with discharge data. We think this are valid justifications to extend the regional record beyond 1971, otherwise we still could use the Eagle data from 1978-2016 (or 1973-2016) to standardize with other regional basins.

Line 252: linear regression models. "simple linear regression" is used to model the relationship between varve thickness and hydrometric variables. Adjusted R2 is listed as an evaluative statistic. Adjusted R2 should be reserved for multiple regression, since it adjusts the coefficient based on the number of independent variables. With only one independent variable, the unadjusted R2 is appropriate (listed as Multiple R-squared in R). Similar with Figs 8 & 9.

Reply: The unadjusted R2 will be used instead of the adjusted R2 as the linear regression model implied only one independent variable.

Line 371-374: This triggered a flag for me – why did the 1971 changes result in a thick and coarse unit? It is explained later on (section 5.2) but left me wanting more explanation here in the results section.

Reply: We agree that it would be useful to include in the revised version of the manuscript a short explanation why did the 1971 changes result in a thick and coarse in this section of the text.

Line 411-: a lot of p-values shown here using a 0.05 threshold (and Table 1). This defeats the purpose of using p-values which are intended to show the actual probability of attaining the particular statistic. Really this is just the same as accept or reject at 95% confidence, which is far too arbitrary. Can these threshold values be replaced with actual p-values to make the analysis more objective? To make matters worse, the threshold value changes to 0.01 in Fig 6. Reporting actual p-values will help with consistency. In line 435-438 there are several r values with no p-value attached. They are "significant" correlations, but no indication of how significant. I would suggest actual

p-values to 3 decimal places would suffice.

Reply: We agree with that comment. P-values shown in the manuscript using threshold values (0.05 and 0.01) will be replaced with actual p-values to make the analysis more consistent and objective.

Line 474: "1972 is considered as an outlier". Is this a subjective consideration or is it supported by the statistical analyses? For example, does the leverage for 1972 appear high when evaluating the regression analyses?

Reply: The fact that the varve of the year 1972 is considered as an outlier is supported by the statistical analyses. When evaluating the regression analyzes, 1972 is far high from the cloud. Indeed, this lamination is interpreted as not being caused by natural hydrological conditions but rather by anthropogenic modification of the watershed. The thickness and the grain size from this varve don't match the annual hydrological instrumental data. Adding 1972 would have the effect of changing the position of the least squares line and inducing an error in the linear regression between variables.

Line 32: take[s]

Reply: OK

Line 69: method[s]

Reply: OK

Line 79: switch "into" and "the" around

Reply: OK

Line 135: [a]eolian [this is very picky]

Reply: OK

Line 157: [an] undisturbed or undisturbed area[s]

Reply: OK

Line 211: Using [a] custom

Reply: OK

Line 227: replace indice with index

Reply: OK

Line 244: allows [an extension to the] instrumental

Reply: OK

Line 249 Table 2: km2 - add superscript

Reply: OK

Line 255: Model[s]

Reply: OK

Line 275: station[s]

Reply: OK

Line 279: "thanks to the. . ." – this is rather informal compared to the rest of the writing. Change to "using the Oudin et al. . ."? Same on line 304.

Reply: OK

Line 378-379: structures allowed [to build] a robust age-model reproducible among cores [to be constructed].

Reply: OK

Line 379: why is the 1 – 5 km distance "significant"? Significant with respect to what? Suggest removing the word.

Reply: OK

Line 392: ([F]ig. 6a)

Reply: OK

Line 401/415: "slight" – what does this mean? Can this decrease in TVT/DLT be supported statistically?

Reply: We will support this decrease statistically in the revised mansucript.

Line 444: [since]

Reply: OK

Line 490: 1887-1991 – should this be 1887-1891?

Reply: Yes, this will be changed.

Line 491-493: this sentence is incomplete. Perhaps solved by removing the "While" at the beginning.

Reply: OK

Line 500: varve[s]

Reply: OK

Line 514: replace on with for

Reply: OK

Line 538/589: important. What does this mean? It seems to be used as a synonym for significant, but it doesn't fit well. The sentences work without the adjective.

Reply: OK

Line 552: Beaver[s]

Reply: OK

Line 583: "floods of [the years] 1972 CE [has (have)] remobilized"

Reply: OK

Line 588: bank[s]

Reply: OK

Line 589: [r]iver

Reply: OK

Line 595: replace for with to

Reply: OK

Line 625: good – another of those pesky vaguely meaningful words. What does it mean in this case – what is a good correlation? Can 'significant' be used here instead? Line 634: global. Do these cores contain a global hydro-climatic signal? Or is it regional (see line 92)?

Reply: That will be clarified.

Line 685: recorded in [the] Grand Lake. . .

Reply: OK

Line 699: discharge[s]

Reply: OK

Line 746: record[s]

Reply: OK

---

## Author Response (AR1)

**cp-2020-87**

**Reconstructing past hydrology of eastern Canadian boreal catchments using clastic varved sediments and hydro-climatic modelling: 160 years of fluvial inflows**

By Antoine Gagnon-Poiré, Pierre Brigode, Pierre Francus, David Fortin, Patrick Lajeunesse, Hugues Dorion and Annie-Pier Trottier.

**Point-by-point reply to the three referee comments**

The review made by the three referees is gratefully acknowledged. The manuscript has been modified in response to the comments and suggestions. In the following document, the referee comments are listed in italic blue, the specific responses to referee comments previously given by the authors during the public discussion are in italic black, and the modifications of the manuscript made by the authors are in black. The line numbers mentioned in section "authors' modification", refer to the line numbers in the Marked-up manuscript. The marked-up manuscript version is presented at the end of this document.

**REFEREE #1**

*This paper presents a short varve chronology from Labrador along with various hydroclimatic interpretations. The identification of varves in this particular region is important due to the limited availability/identification of palaeoenvironmental proxies in the Boreal region of eastern Canada. This work proposes to help fill that gap. The authors suggest a potential for a longer-term record to emerge from this lake – this would greatly benefit hydro-climate reconstructions in this region. The palaeohydrologic interpretation of the varve record is robust, supported by independent dating and multiple statistical approaches.*

*Overall, the sedimentary analyses and interpretations are sound. Most of my comments below focus on the reporting of the statistical analyses. The figures are well drawn. There is a heavy reliance on acronyms which take some time to get familiar with. In many places, a comma is used instead of a period for quantities (eg, Fig 9 vs Table 1) – from a Canadian perspective it doesn't matter which is used but pick one convention for consistency. Four research objectives are identified in the introduction, and the paper discusses each of these sufficiently.*

*I would recommend publication of this manuscript with the below comments/suggestions/questions addressed.*

> *Reply:*
>
> *We thank reviewer #1 for his positive comments on our manuscript. The use of acronyms reduced to facilitate the reading of the manuscript. Also, a descriptive table grouping the main acronyms used in this study could be a good way to help readers familiarised with acronyms and make the reading more fluid. The used of the period for quantities uniformized.*
>
> Authors' modification:
>
> The acronyms used for the 3 seasonal sub-layers (ESL, DL and AWL) are no longer used in the text. Same for PSI (particle-size indices) and IRD (ice-rafted debris). The acronyms defining varve's physical parameters (TVT, DLT and $P99D_0$) are still present in the text considering they are widely seen in literature on varved sediment.

**Specific comments:**
*Line 111-113: how are "winters" and "summers" defined? Later in the paragraph the snowmelt season is defined as AMJ, but there is no similar definition of the seasons. Assume JFM and JAS?*

> *Reply:*
>
> *Winter (DJFM) and summer (JJA) will be defined in the revised version.*

Authors' modification:

Winter (DJFM) and summer (JJA) are defined, line 134.

*Line 189: "counts were executed repeatedly". How were the counts made? Multiple counters? Multiple counts per counter? There is a mention of counting difficulty (line 382). If multiple counts were made, how consistent were those counts? Given the clear images and laminae it would seem to be fairly clear-cut, but I'd like to see some mention of the accuracy/precision of the counting process to fortify that.*

Reply:

*Due to the great quality of the varved sequences, two counts were made by one counter (AGP). As mentioned in the text, counting difficulties occur within varve years 1952-1953, 1935-1934, 1918-1919. The error percentage between the 2 counts will be mentioned in the text to further demonstrate the accuracy/precision of the counting process.*

Authors' modification:

The counting error percentage is now mentioned, line 521.

*Line 244-245: Only 1 of the 5 instrumental records goes back to 1966 (incomplete data 1966-68?). Is this good enough to extend the composite instrumental record back to 1969? "Strong positive correlations" are stated but not shown – could these be added to Table 2? Also, the extension crosses pre- and post-diversion boundary – is it still reasonable extend the record back past 1971?*

Reply:

*There is an error in the Tab. 2, the Eagle series goes from 1969 to 2016, not 1966 to 2016. This will be corrected. Dinis et al., 2019, produced an observed river index of summer regional discharge in Labrador for the 1969-2009 period. They normalised and average hydrometric data from the Eagle River; 1969–2009, Alexis River; 1978–2009 and Little Mecatina River; 1979–2009. As this paper has been accepted and published in the international journal Climate Dynamics, we think it is reasonable to use the same methodology to produce our Labrador region mean annual discharge series.*

*Dinis, L., Bégin, C., Savard, M. M., Marion, J., Brigode, P., and Alvarez, C.: Tree-ring stable isotopes for regional discharge reconstruction in eastern Labrador and teleconnection with the Arctic Oscillation, Clim. Dynam., 53, 3625-3640, https://doi.org/10.1007/s00382-019-04731-2, 2019.*

*The significant positive correlations between the four streamflow series (Tab. 2) with Naskaupi River discharge mentioned in the section '' 3.4 Hydro-climatic variables used'' (line 242) are rather shown in the result section ''4.5 Relation*

*between varve series and instrumental record" (line 455) because we believe that these are results. A note in section 3.4 will be added to guide readers (i.e. see section 4.5 for details on correlations).*

*We think that it is useful to extend the regional mean annual discharge series back past diversion boundary with the Eagle River hydrometric data because this produce a longer calibration period using a large watershed of the Labrador region which is devoid of anthropogenic modifications. This also allows to calibrate the few varves pre-diversion with discharge data. We think this are valid justifications to extend the regional record beyond 1971, otherwise we still could use the Eagle data from 1978-2016 (or 1973-2016) to standardize with other regional basins.*

Authors' modification:

Discussion on the regional hydrological signal in Labrador and the similarities between the Naskaupi River hydro-climatic variables with other Labrador hydrometric stations is now presented in section 4.5, line 638.

The period covered by the Eagle hydrometric station instrumental data (1969 to 2011) has been corrected in Tab. 2.

*Line 252: linear regression models. "simple linear regression" is used to model the relationship between varve thickness and hydrometric variables. Adjusted R2 is listed as an evaluative statistic. Adjusted R2 should be reserved for multiple regression, since it adjusts the coefficient based on the number of independent variables. With only one independent variable, the unadjusted R2 is appropriate (listed as Multiple R-squared in R). Similar with Figs 8 & 9.*

> *Reply:*
>
> *The unadjusted $R^2$ will be used instead of the adjusted R2 as the linear regression model implied only one independent variable.*

Authors' modification:

Adj $R^2$ was changed for $R^2$ in the text and figures.

*Line 371-374: This triggered a flag for me – why did the 1971 changes result in a thick and coarse unit? It is explained later on (section 5.2) but left me wanting more explanation here in the results section.*

> *Reply:*
>
> *We agree that it would be useful to include in the revised version of the manuscript a short explanation why did the 1971 changes result in a thick and coarse in this section of the text.*

Authors' modification:

We clarified the short text on the suggested link between the distinct 1972 marker layer with the occurrence of the Naskaupi River diversion, which supports the reliability of the constructed chronologies (line 507). We also refer the readers to section 5.2 for more details.

*Line 411-: a lot of p-values shown here using a 0.05 threshold (and Table 1). This defeats the purpose of using p-values which are intended to show the actual probability of attaining the particular statistic. Really this is just the same as accept or reject at 95% confidence, which is far too arbitrary. Can these threshold values be replaced with actual p-values to make the analysis more objective? To make matters worse, the threshold value changes to 0.01 in Fig 6. Reporting actual p-values will help with consistency. In line 435-438 there are several r values with no p-value attached. They are "significant" correlations, but no indication of how significant. I would suggest actual p-values to 3 decimal places would suffice.*

> *Reply:*
>
> *We agree with that comment. P-values shown in the manuscript using threshold values (0.05 and 0.01) will be replaced with actual p-values to make the analysis more consistent and objective.*

Authors' modification:

P-values for the main correlations are now shown in Fig. 8, 9, 10 of the manuscript and in Tab. S4 and S5 of the Supplements.

*Line 474: "1972 is considered as an outlier". Is this a subjective consideration or is it supported by the statistical analyses? For example, does the leverage for 1972 appear high when evaluating the regression analyses?*

> *Reply:*
>
> *The fact that the varve of the year 1972 is considered as an outlier is supported by the statistical analyses. When evaluating the regression analyzes, 1972 is far high from the cloud. Indeed, this lamination is interpreted as not being caused by natural hydrological conditions but rather by anthropogenic modification of the watershed. The thickness and the grain size from this varve don't match the annual hydrological instrumental data. Adding 1972 would have the effect of changing the position of the least squares line and inducing an error in the linear regression between variables.*

Authors' additional reply:

Here is the scatter plot presented in Fig. 9 including the 1972 detrital layer thickness measurement. The 1972 detrital layer thickness, which is far from the cloud, is considered as an outlier and was not included in all reconstructions.

[Figure]

**Technical corrections: remove/add what is in [ ]**

*Line 32: take[s]*

> *Reply:*
> *OK*

> Authors' modification:
> Done, line 32.

*Line 69: method[s]*

> *Reply:*
> *OK*

> Authors' modification:
> Done, line 73.

*Line 79: switch "into" and "the" around*

> *Reply:*
> *OK*

> Authors' modification:
> Done, line 95.

*Line 135: [a]eolian [this is very picky]*

> *Reply:*
> *OK*

> Authors' modification:
> Done, line 160.

*Line 157: [an] undisturbed or undisturbed area[s]*

> *Reply:*
> *OK*

> Authors' modification:
> Done, line 184.

*Line 211: Using [a] custom*

> *Reply:*
> *OK*

> Authors' modification:
> Done, line 278.

*Line 227: replace indice with index*

> *Reply:*
> *OK*

> Authors' modification:
> Indice was replaced by ''The $99^{th}$ percentile'', line 294.

*Line 244: allows [an extension to the] instrumental*

> *Reply:*
> *OK*

> Authors' modification:
> Done, line 317.

*Line 249 Table 2: km2 - add superscript*

> *Reply:*
> *OK*

> Authors' modification:
> Done, line 322.

*Line 255: Model[s]*

> *Reply:*

> *OK*

> Authors' modification:

> Done, line 337.

*Line 275: station[s]*

> *Reply:*

> *OK*

> Authors' modification:

> This text was removed, line 357.

*Line 279: "thanks to the. . ." – this is rather informal compared to the rest of the writing. Change to "using the Oudin et al. . ."? Same on line 304.*

> *Reply:*

> *OK*

> Authors' modification:

> Done, line 362.

*Line 378-379: structures allowed [to build] a robust age-model reproducible among cores [to be constructed].*

> *Reply:*

> *OK*

> Authors' modification:

> Done, line 514.

*Line 379: why is the 1 – 5 km distance "significant"? Significant with respect to what? Suggest removing the word.*

> *Reply:*

> *OK*

> Authors' modification:

> Done, line 515.

*Line 392: ([F]ig. 6a)*

    *Reply:*

    *OK*

    Authors' modification:

    Done, line 538.

*Line 401/415: "slight" – what does this mean? Can this decrease in TVT/DLT be supported statistically?*

    *Reply:*

    *We will support this decrease statistically in the revised manuscript.*

    Authors' modification:

    Statistical support is now available in the Supplements Fig. S1, S2, S3 and Tab. S1, S2, S3.

*Line 444: [since]*

    *Reply:*

    *OK*

    Authors' modification:

    This text was removed, line 694.

*Line 490: 1887-1991 – should this be 1887-1891?*

    *Reply:*

    *Yes, this will be changed.*

    Authors' modification:

    Done, line 740.

*Line 491-493: this sentence is incomplete. Perhaps solved by removing the "While" at the beginning.*

    *Reply:*

    *OK*

    Authors' modification:

    This sentence has been removed from this section, line 741.

*Line 500: varve[s]*

> *Reply:*
>
> *OK*

> Authors' modification:
>
> This text has been modified, line 793.

*Line 514: replace on with for*

> *Reply:*
>
> *OK*

> Authors' modification:
>
> Done, line 819.

*Line 538/589: important. What does this mean? It seems to be used as a synonym for significant, but it doesn't fit well. The sentences work without the adjective.*

> *Reply:*
>
> *OK*

> Authors' modification:
>
> "important" was removed, line 834.

*Line 552: Beaver[s]*

> *Reply:*
>
> *OK*

> Authors' modification:
>
> Done, line 849.

*Line 583: "floods of [the years] 1972 CE [has (have)] remobilized"*

> *Reply:*
>
> *OK*

> Authors' modification:
>
> Done, line 919.

*Line 588: bank[s]*

> *Reply:*
>
> *OK*

Authors' modification:

Done, line 917.

*Line 589: [r]iver*

*Reply:*

*OK*

Authors' modification:

Done, line 918.

*Line 595: replace for with to*

*Reply:*

*OK*

Authors' modification:

Done, line 955.

*Line 625: good – another of those pesky vaguely meaningful words. What does it mean in this case – what is a good correlation? Can 'significant' be used here instead? Line 634: global. Do these cores contain a global hydro-climatic signal? Or is it regional (see line 92)?*

*Reply:*

*That will be clarified.*

Authors' modification:

"Significant" was used instead of "good", line 999.

*Line 685: recorded in [the] Grand Lake. . .*

*Reply:*

*OK*

Authors' modification:

Done, line 1126.

*Line 699: discharge[s]*

    *Reply:*

    *OK*

    Authors' modification:

    Done, line 1141.

*Line 746: record[s]*

    *Reply:*

    *OK*

    Authors' modification:

    ''record'' has been replaced by ''sequence'', line 1229.

**REFEREE #2**

**General comments:**

*The manuscript by Gagnon-Poire and co-authors entitled 'Reconstructing past hydrology of eastern Canadian boreal catchments using clastic varved sediments and hydro-climatic modeling: 160 years of fluvial inflows' presents river discharge reconstructions from three short cores containing clastic varves reaching 160 years back in time. For the discharge reconstruction mainly two proxies have been applied (grain size and layer thickness). These data demonstrate the large potential for discharge reconstructions using annually laminated sediments.*

> Reply:
>
> *We thank reviewer #2 for his positive comments on our manuscript.*

*However, a few week points in the interpretation need to be better clarified. In general, it is difficult to follow the large number of different statistical correlations between cores, proxies, proxy reconstruction and model results. A more concise approach with a focus on main correlations would make the manuscript easier to read. Furthermore, instead of levelling out the different signals in the three cores by a pooling approach, the causes for these differences should be better examined and documented. The implications of the difference between cores for selecting the most suitable core location for palaeoydrological reconstruction should be elaborated.*

> Reply:
>
> *We have indeed tried to reconstruct streamflow using single core data and all possible core combinations. However, statistical analysis of these reconstructions shows poorer results (un-significant p values, negative average reduction of error (RE) and negative average coefficient of efficiency (CE) (values > 0 are needed to validate the twofold cross-validation technique). The pooled data from the 3 cores (mean DLT series and mean $P99D_0$) are the combinations showing the best statistical results (calibration and validation).*
>
> *We used pooled data from 3 cores in order to better capture the regional hydroclimatatic data, and also to somehow remove the noise that is inherent from the analysis of the tiny part of a single core in a very large lake. We do not believe that selecting a single "most suitable core" for paleohydrological reconstruction is the right strategy because a single core will be more sensitive to local disturbances and is probably less representative of the entire hydrogram.*
>
> *One of the main goals of the paper is making the demonstration that Grand Lake sediments record a regional hydroclimate signal, not only to reconstruct the Naskaupi river hydrogram. We will clarify this in the revised version of the manuscript. Nevertheless, we agree it would be useful to include in the revised*

*version of the manuscript a better explanation of the causes of the differences between the cores.*

Authors' modification:

Text was added in section 5.1. line 853, and 5.3, line 1009, to better justify the use of the combined series from the 3 sites for reconstructions, which in our opinion, allows to better capture the hydrological signal from a larger region.

Section 4.6 has been modified to better support the choice of the proposed reconstructions. For the sake of transparency, all the other reconstrcutions, i.e. Naskaupi River Q-mean and Q-max reconstructed from the DLT and $P99D_0$ series using single-core data, the combined series, and other core combinations are now presented in the Supplements Fig. S4 and S5. Results of the model calibration for all Naskaupi River Q-mean, Q-max and Labrador region Q-mean reconstructions are also presented in the Supplements Tab S6, S7, S8, S9, S10, S11. We decided to include these informations in Supplements in order to keep the manuscript as simple as possible, as suggested by the reviewer.

The "mean DLT and $P99D_0$ series" used to define the pooled data from the 3 sites in the previous version of the manuscript are now named "combined DLT and $P99D_0$ series". Data from the three sites have been normalized and averaged to produce combined series.

*The cores have been taken from different parts of the delta surface and even the most distal core location is still 70 m above the deep basin. Sediment reworking processes on the delta should have an influence on the deposition and layer thickness as well as grain size. For example, a thinning of discharge layers from the proximal to the distal delta location (NAS-1 to NAS-2) should be expected, which, however, is not seen in the layer thickness plots shown in figure 6. A more detailed discussion of sedimentological processes on the delta surface should be added for clarification.*

*Reply:*

*Thank you for this suggestion, we will add a discussion about the sedimentological processes on the delta surface, although core NAS-2 is no longer on the delta itself. We will locate the NAS-1 coring site on the 3.5 kHz subbottom profile of the Naskaupi River delta on the Fig. 1C to help visualize the Naskaupi deltaic context and feed the discussion on sedimentological processes. Yet, there is a thinning of the detrital/discharge layers between NAS-1 and NAS-2, although quite small indeed. The mean DTL thickness of both cores will be added. It is clearly visible on Figure 4. In the context of this very large lake, the distance between the 2 cores is quite small, so we are not surprised to see such a small difference, especially considering that the laminations are still formed at*

*the very end of the end (+/- 45 km away) and can be correlated with laminations from the proximal zone. The grain size is also finer in NAS-2 compared to NAS-1. The median grains size of both cores will be added.*

Authors' modification:

Similarities of sedimentological processes between sites is now discussed in section 5.1, line 853. Specifications on the coring site's location was added in section 3.1, line 185. The location of the NAS-1 site is shown on the 3.5 kHz subbottom profile of the Naskaupi River delta on the Fig. 1c. Note that the vertical exaggeration is 12x. This helps to visualize that the NAS-1 site is located in a relatively flat area where reworking processes are not conducive. The mean of the thickness and particle size measurements are presented in the Supplement's Tab. S1, S2, S3, showing that their absolute values are in the same range, indicating that they are likely produced by similar sedimentary processes.

*The 'anthropogenic impact' after dyke construction (in 1971 or 1972?) has been stressed several times (e.g. lines 444/445). However, it is not clear how exactly dyke construction impacted on the sedimentation. Was the main effect generated by the earth movements during dyke construction (if at all, how long did his effect last?) or by the reduction of the catchment? If dyke construction resulted in 'increased availability of sediments in the river system' as suggested (lines 588-589), why is that only seen in NAS-1 core? Why should there be more sediments in the system although the catchment size decreased? The different behavior of the cores NAS-1 and NAS-2 after 1972need to be better elaborated. The argumentation that NAS-2 behaves like BEA-1 (lines598 and following) is not convincing because the BEA-1 location is not affected by the Naskaupi River inflow, whereas NAS-1 and NAS-2 are located in the same direction towards the river inflow. Furthermore, in contrast to DLT, grain-size data do not show major difference between both cores after 1972. How is that explained?*

*Reply:*
*We are quite surprised by these comments. Section 5.2 answers most of these questions: for instance, the reviewer question, "Why should there be more sediments in the system although the catchment size decreased?", was answered in lines 585-589: "The reduction of nearly half of the area of the Naskaupi River watershed reduced the water inflows and changed the base level of the downstream river system. The rapid base level fall must have triggered modifications of the fluvial dynamics such as channel incision, banks destabilization and upstream knickpoint migration, likely increasing the availability of sediments in the River system.". Maybe the arguments were not enough clearly outlined, and we will make sure to improve the clarity of that section. What is certain is that the varve structure in both NAS-1 and NAS-2*

*cores changed after 1972, and we will emphasize that feature in the revised version of this section 5.2.*

Authors' modification:

Section 5.2 has been modified to better explain the effect of the dyke system on the Naskaupi River sediment inputs.

*Due to the core differences, post 1972 DLT data of NAS-1 were excluded from statistical analyses? Instead of excluding the data, correlation of NAS-1 and NAS-2 core data post 1972 with hydrological data should be compared. It would be interesting to see how the sedimentological differences affected the correlations with hydrological data.*

> *Reply:*
>
> *As mentioned earlier, we tried to reconstruct streamflow using single core data and all possible core combinations. Maybe could we outline this in the supplementary data in order to keep the manuscript as simple as possible, focusing on the main arguments as suggested by the reviewer.*

Authors' modification:

The combined DLT series without the 1972-2016 period presents a slightly better fit with the instrumental data (Supplement Tab. S6, S7). However, there are small differences between reconstructions using the combined DLT and P99D$_0$ series and the combined series without the NAS-1 1978-2016 period (Supplement Fig. S4, S5).

*The proxy data from different cores have been pooled to obtain a better statistical correlation with hydrological variables (lines 630-631). However, pooling masks the different sensitivity of the different core locations in recording natural hydrologicial variability. Moreover, it is not clear if the pooling includes all data from all cores or if some parts of the data are excluded. In line 614 it was pointed out that the post 1972 period has been excluded from one of the cores (NAS-1). If this part of the record is also not included in the pooling approach you put apple and pears in the basket and I wonder about the meaning of improved statistical correlation. Since the BEA-1 and NAS-1 (lines 599-604) are considered to record the 'natural hydro-climatic signal' one should expect a better representation of palaeohydrogical changes in one of these cores rather than in pooled data from all cores.*

> *Reply:*
>
> *Well, our text in lines 599-604 explains that BEA-1 and NAS-2 (not NAS-1) are considered to record the 'natural hydro-climatic signal', i.e. without the influence of the dyke. So maybe there is some sort of misunderstanding here.*

Authors' modification:

As mentioned above, section 4.6 has been modified to better justify the proposed reconstructions in the manuscript. Text was added in the section 5.3 (line 1009) to better justify the use of the combined series from the 3 sites for reconstructions.

*The authors report variability on different time scales, i.e. long-term trends in mean annual discharge (line 687) and decadal-scale variability (e.g. lines 56-57) but they do not explicitly relate these. The appearance of variability at different time scales is an interesting finding that should be more emphasized and elaborated in the paper.*

*Reply:*

*Yes indeed, this is an interesting finding, but this theme will be exploited in an upcoming paper from the same site with a longer and even more interesting record. Unless the editor wants us to expand on this, we would like to hold that information for the time being.*

Authors' modification:

This theme will be discussed in detail in an upcoming article from the same site. The observation of variability on different time scales is no longer reported in the manuscript. The following text discussing this aspect has been removed from the manuscript: "Reconstructed Q-max series reveals more significant interannual and decadal-scale variability, however long-term trend is observed in both reconstructed Q-mean and Q-max series. Ongoing work on Grand Lake varved sequence suggests that variability in river discharge may occur at different timescales in the Labrador region".

*The statement about dyke effects on sediment transport and its 'implications for palaeohydrological reconstruction' (lines 703-705) and that dyking effects are 'clearly visible in the sedimentary record' (lines 743-744) are too much simplified. It has been shown that one coring sites has been affected by dyke construction but the two others not or only to a minor degree. This differentiation between core locations is an important point and knowledge about these differences and their causes is essential to select the most suitable coring locations for palaeohydrological reconstruction. In this respect, and here I repeat my previous comment, I do not consider the pooling as suitable approach even if it may improve statistical correlation. Often unspecific terminology is used like, for example, 'thick and coarse', 'thicker' (examples in specific comments). This should be changed into quantified information.*

*Reply:*

*We agree to improve the text related to the explanation of the dyking effects, and augment our discussion about the differences in sedimentary processes occurring in the coring sites. We will make our terminology more specific, and change it in quantified information.*

Authors' modification:

As mentioned above, section 5.2 has been modified to better explain the effect of the dyke system on the Naskaupi River sediment inputs. Similarities of sedimentological processes between sites are now discussed in section 5.1, line 853.

The unspecific terminology was changed into quantitative data. Additional visual support and statistical information on TVT, DLT and $P99D_0$ series from each different core are now provided in the Supplement.

**Specific comments:**

*A number of 'distinctive marker layers' (labelled A-P, Figure 4, lines 381, 382) have been defined but it is not explained how distinctive these layers are and what makes them distinctive. In figure 4 they do not appear distinctly different neither in the core image nor in the XRF data.*

*Reply:*

*An explanation will be added.*

Authors' modification:

Chronology of each core was confirmed by cross-correlation between thick laminations selected as distinctive marker layers along the different sediment sequences (section 3.2, line 239).

*In the chapter 'Regional setting' some information about vegetation cover should be added since that may influence catchment erosion and clastic sediment transport into the lake.*

*Reply:*

*We will specify what is the vegetation of the High Boreal Forest ecoregion.*

Authors' modification:

Specifications on vegetation cover was added in section 2, line 131 and 137.

*In chapter 4.7 it is not clear which sediment proxies have been compared with the rain fall-runoff modeling approach. Are these proxy data from individual cores (which?) or from pooled data? If it is pooled data, how did you account for differences in TVT between cores?*

*Reply:*

*We will specify that it is from pooled data, and we will provide the comparison for each core in a supplement in order to keep the MS simple.*

Authors' modification:

We specified that it is the combined DLT and $P99D_0$ that have been compared with the rain-fall runoff modeling approach (section 4.7, Fig. 10 caption).

*Line 162: It should be specified which efforts were made to retrieve undisturbed sediment surfaces. Taking short cores from such deep lakes without disturbance is a common problem to the community and it would be helpful to know how the authors tried to improve the coring in this respect.*

 *Reply:*

 *This will be specified.*

 Authors' modification:

 Our technique to retrieve undisturbed sediment surfaces is now explained in section 3.1, line 191.

*Lines 185-186: Sampling intervals for Cs-dating are unclear. Was it attempted to sample individual varves or only sublayers? Sample intervals vary between 2 and 0.5 cm but according to figure 6 layer thickness was > 4cm? Please clarify.*

 *Reply:*

 *This section is confusing and will be clarified.*

 Authors' modification:

 Description of the sampling intervals for Cs-dating has been clarified (section 3.2, line 230).

*Line 226: Specify 'coarse debris' and quantify grain sizes*

 *Reply:*

 *This will be done.*

 Authors' modification:

 Grain size of the coarse debris observed in the early spring layer (μm to mm scale) is mentioned in section 4.1, line 424.

*Line 227: Explain the PSI. Is this a mean grain size for each lamination? What is 'lamination' in this respect? A varve or a sublayer (which?)?*

 *Reply:*

 *This will be done.*

The Acronym PSI (particle-size indices) is no longer used in the text and have been replaced by the 99th percentile ($P99D_0$) of the particle size distribution. The $P99D_0$ was obtained for each detrital layer (section 3.3, line 295).

*Line 325: What is 'occasionally'? Provide the number or percentage of DL with sharp lower boundary.*

> *Reply:*
>
> *This information will be added.*

Authors' modification:

"Occasionally" has been removed. The detrital layer always has a sharps lower boundary (section 4.1, line 428).

*Line 327: Explain 'non-annual' for these layers. All three described sub-layers (ESL, DL, AWL) are seasonal, i.e. non-annual. Also quantify 'thin coarser'. What is the thickness (range or mean) and grain size of these layers? Finally, quantify 'some cases',i.e. how many of these layers did you count?*

> *Reply:*
>
> *This will be explained.*

Authors' modification:

The term "Non-annual" is no longer used. The varve structure can be divided in 3 seasonal layers. So, now we say: the upper part of the detrital layer consists of a finer detrital grain matrix containing thin visually coarser intercalated sub-layers in ~75% of the laminations (line 429). However, we did not calculate the particle size of these particular sub-layers individually.

*Lines 328-329: Provide information why Ca and Sr are relatively higher in DLs, i.e. which minerals in the DLs include these elements?*

> *Reply:*
>
> *Allochthonous lithoclastic materials that composed the DLs are rich in Ca and Sr. These elements come mainly from eroded sediments of the Grenville geological province (i.e. plagioclase, granodiorite?) deposited in the Grand Lake's watershed during glacio-marine/lacustrine phase and remobilized by spring floods. We did not perform EDS analysis.*

Authors' modification:

We now say to be more specific that: "The allochthonous lithoclastic materials which compose the detrital layers are associated with higher density values (Fig.

4) and an increase in the relative intensity of elements Sr and Ca (Zolitschka et al., 2015)" (section 4.1, line 430).

*Line 344: 'thick and coarse' is unspecific. Provide information about thickness and grain size of this prominent layer. Are there distinct differences also in the elemental composition of this layer?*

> *Reply:*
>
> *This section will be clarified.*

Authors' modification:

Thickness and grain size of the 1972 marker layer were provided in section 4.1, line 467.

*Lines 349/350/351: the ESL of pre-1972 CE is 'thicker'. Provide quatified information instead of this unspecific information. It should be easy to calculate mean contribution of the ESL (in %) to the total varve thickness for the pre- and post-1972 intervals*

> *Reply:*
>
> *This will be done.*

Authors' modification:

The mean contribution of the early spring layer and autumn and winter layer to the total varve thickness for the pre- and post-1972 intervals are provided in section 4.1, line 475.

*Lines 350, 352: 'post-1971' or 'post-1972'?*

> *Reply:*
>
> *This will be clarified.*

Authors' modification:

Post-1972, i.e. after the Naskaupi River diversion effect (line 470 and 473).

*Lines 372/373: When exactly was the anthropogenic change in the catchment? Was it in the year before the 1972 marker layer or in 1972? If it was in the year before, why was there a 1 years delay in the sediment response?*

> *Reply:*
>
> *On 28 April 1971, by closing a system of dykes, the headwaters of Naskaupi River watershed were diverted into the Churchill River hydropower development. The base level fall must have triggered modifications of the fluvial dynamics such as channel incision, bank destabilization and upstream knickpoint migration*

*during the rest of the year. We interpret that it was only during the following spring flood (1972) that the destabilized sediments (during the previous year) were the most remobilized and deposited on the Naskaupi delta. This section will be clarified.*

Authors' modification:

The exact date of the dyke construction (April 1971) is now mentioned in section 5.2, line 914. The text has been reworked to better explain when the anthropogenic change occurred in the catchment and the 1-year delay in the sediment response.

*Figure 6. Add the position of marker layers A-P in the figure.*

*Reply:*

*This will be done.*

Authors' modification:

The position of marker layers A to M were added in the Fig. 6.

*Lines 414 and following: How is the P99D0 value influenced by the ratio DL/TVT?*

*Reply:*

*There is a significant positive correlation ($R^2$ = 0.38 p-values = 0.01) between DL/TVT and $P99D_0$. A lamination with a high LDL / TVT ratio is more likely to have high grain size values. However, this correlation shows that DLT and $P99D_0$ remain independent variables and can both reveal different information (i.e. Q-mean and Q-max).*

Authors' modification:

Additional discussion on the relation between TVT, DLT and $P99D_0$ was added in section 4.3, line 599.

*Line 550: How often is 'seldom'? In how many layers erosion traces have been observed.*

*Reply:*

*This will be clarified.*

Authors' modification:

The wording of the original sentence was clumsy. We meant that clasts of eroded material could be found in the early spring layers at 3 sites we sampled, but not that the early spring layers were impacted by erosion at these 3 sites. We modified the text accordingly (section 5.1, line 846 and 853).

*Line 550/551: What kind of traces of erosion are these. Provide a description. I would expect differences between the proximal and distal cores. Please clarify.*

Reply:

*This will be clarified.*

Authors' modification:

This is now explained in section 5.1, line 846 and 853.

*Line 580: I disagree that river sediment input was 'quantitatively and spatially constant' before 1971. There is distinct variability at different time scales in the data, e.g.between 1920 and 1960s.*

Reply:

*Reviewer is right, this statement is confusing, we will be more specific.*

Authors' modification:

This sentence was removed from the text (line 913).

*Line 602-604: It is assumed that 'natural hydro-climatic signal' drives the sedimentation in BEA-1 (and NAS-2) without saying what this 'natural hydro-climatic signal' is. This statement should be easy to be proven or disproven by correlation with instrumental hydrological data.*

Reply:

*This will be done.*

Authors' modification:

Indeed, the observed Naskaupi River Q-mean series also shows a decrease on the 1978-2011 period (section 5.2, line 963).

*Line 634: You will get at best a regional hydro-climatic signal but certainly no global.*

Reply:

*Yes, reviewer is right, that will be changed.*

Authors' modification:

"global" was replaced by "from a larger region", line 1018.

*Line 642: Quantify 'slight variability'*

*Reply:*

*The variability will be quantified.*

Authors' modification:

"slight variability" was removed. This part of the text was reworked (section 5.3, line 1059).

*Line 648: How do you explain 'high thickness values' (need to be quantified!) of ESL sand AWLs during the 1920s?*

*Reply:*

*This will be quantified. Hypotheses will be provided.*

Authors' modification:

This text was removed to keep the manuscript as simple as possible (section 5.3, line 1059), focusing on the main arguments, as suggested by the reviewer.

*Lines 675-677: There is a detailed discussion on thresholds and flood amplitude re-construction in Kaempf et al., 2014 (J. Quat. Sci.) that you may consider including in this part of the discussion.*

*Reply:*

*We are going to consider including this information.*

Authors' modification:

We included this reference in the text (line 1115).

**Technical corrections:**

*Lines 328-329: 'abundance of elements'. This is wrong because XRF scanner data are relative variations of element intensities but not quantified amounts*

*Reply:*

*OK*

Authors' modification:

"abundance of elements" was removed. We now said to be more specific that: "Elements were normalized by the total count (cps) for each spectrum" (line 209).

*Line 547: instead of 'underlying' it should be 'overlying'*

*Reply:*

*OK*

Authors' modification:

Done line 843.

*Line 571 (figure caption): see comment above, XRF data does not give 'abundances'.*
*This are relative changes of element intensities*

*Reply:*

*OK*

Authors' modification:

"abundance" was changed for "relative intensities" in Fig 4. 5 and 11 captions.

**REFEREE #3**

**General comments:**
*This study by Antoine Gagnon-Poiré and colleagues entitled "Reconstructing past hydrology of eastern Canadian boreal catchments using clastic varved sediments and hydro-climatic modeling: 160 years of fluvial inflows" presents an interesting counterpart to rainfall-runoff modeling approaches that aim at expanding instrumental streamflow datasets for multi-decadal analysis of hydrological variability. Indeed, this study based on varved sediment sequences aims at producing long river discharge records (>100 years) to support, help refine or contradict paleo-hydrological records offered by the modeling approaches.*

*The strength of this study is clearly provided by the very high-quality analysis of the varve record and the robustness of the sediment chronology. Varve boundaries are clearly defined through high-quality startigraphical analysis combined with CT images and state-of-the-art microscopy-based grain size analysis. Varve counts are consistent between the cores of different locations, and they are supported by independent 137Cs dating. The varve record thus offers an annual view into past changes without chronological constraints, which is a major advantage for developing a proxy-climate or proxy-hydrology models.*

*Varve stratigrahical analysis further allowed to select the best varve parameter (i.e., meaningful season) to compare with hydrological data. The proxy-hydrology correlations have been significantly improved by selecting the thickness of the detrital layer (DLT) instead of total varve thickness (TVT), thus reducing potential noise; spring discharge being the main driver for sediment erosion and transport in the nival catchment of Naskaupi River. In this context, Figure 11 is very stunning, and shows how a varve record can best be exploited to look at micro-meteorology and lower-than-seasonalresolution river hydrodynamics; this is novel.*

*However, although the quality of the sedimentary investigation is very robust, general important comments relate to the methods to produce the paleo-hydrological record and its regional signal. I hope that these major comments will be well received and accepted, and that they will be of good use to improve the present manuscript.*

> Reply:
> We thank reviewer #3 for his positive comments on our manuscript.

*Normalizing total varve thickness (TVT) is interesting when several sediment cores are collected at the same location => thus to reduce local error in the proxy-hydro/climate relationship. However, merging TVT from a proximal (more sensitive, thus with larger amplitude) and distal record (buffering large changes in river discharge, recording annual change in hydrodynamics and only sensitive to the most intense discharge events) is neither properly justified in the text, nor fully appropriate. It gives the impression that the different records were merged in the way that the correlation with*

*hydrometric data would be maximize, at the cost of process understanding. A great example is losing the downward trend in TVT from NAS-2 by merging its record with NAS-1, which has no trend. The same applies to (and I would say particularly applies to) P99D0. Mean values are strongly driven by NAS-1, the proximal coring site. As such, it is not surprising to find the best correlation for Qmax to NAS-1 (proximal) and for Qmean to NAS-2 (distal). Overall, there is no mechanistic logical explanation in merging TVT, DLT or P99D0 from the three cores to help maximize the correlation. This is particularly the case integrating BEA core, for which it is argued (L604) that "it is quite unlikely that the sedimentary input from the Naskaupi River contributed to sediment accumulation at the mouth of the Beaver River" (i.e., BEA core). L443: There is no clear explanation on why the post-anthropogenic watershed modification would support the discarding of NAS-1 in the TVT, DLT and P99D0 normalization of the cores. It further supports the impression that the best records were merged in the way that the correlation with hydrometric data would be maximize, at the cost of process understanding. L461: Table 3 is named Table 1. . ..it took me some time to realize that Table 3 was not missing, while being important and largely cited.*

*Reply:*

*Thank you for that comment. Considering the very large size of the lake, the coring sites are quite close to each other, especially NAS-1 and NAS-2 (~1km). It is more than probable that the sediment deposition phenomena at the different sites are similar. We normalized and pooled data from 3 cores in order to somehow reduce the local sensibility recorded and better capture the regional hydroclimatic signal.*

*We will better explain our choice to merge the DLT or $P99D_0$ of the three cores in the revised version of the manuscript. We will show in supplement material of the revised version streamflow reconstructions using single core data and all other core combinations. This will help discuss process understanding, anthropogenic watershed modification and the result of adding and discarding some cores or core section.*

*We consider that the Naskaupi and the Beaver rivers have a very similar annual hydrological dynamic due to their close proximity. (L604) Evidence leads us to believe that it is quite unlikely that the sedimentary input from the Naskaupi River contributed to sediment accumulation at the mouth of the Beaver River. The BEA core does not record the Naskaupi River input but rather the hydrological conditions of the Beaver River which are quite similar. With the meteorological dataset used in our study (e.i. temperature and precipitation), it appears that the two catchments have very similar climatological characteristics. Integrating BEA core in the pooled data allows to capture the hydrological signal from a larger region (Nakaupi + Beaver watersheds).*

*The mistake concerning the Table 3 will be corrected, sorry for that.*

Authors' modification:

As mentioned in the referee **#**2 response, similarities of sedimentological processes between sites is now discussed in section 5.1, line 853.

Text was added in section 5.1. line 853 and 5.3, line 1009, to better justify the use of the combined series from the 3 sites for reconstructions.

Naskaupi River Q-mean and Q-max reconstructed from the DLT and $P99D_0$ series using single-core data, the combined series, and other core combinations are now presented in the Supplements Fig. S4 and S5. Results of the model calibration for all Naskaupi River Q-mean, Q-max and Labrador region Q-mean reconstructions are also presented in the Supplements Tab S6, S7, S8, S9, S10, S11.

The "mean DLT and $P99D_0$ series" used to define the pooled data from the 3 sites in the previous version of the manuscript are now named "combined DLT and $P99D_0$ series". Data from the three sites have been normalized and averaged to produce combined series.

An explanation on why we suggest discarding of NAS-1 1978-2011 period in the combined DLT series for Q-mean reconstructions is provided in section 4.6, line 703.

Table 3 is now correctly named.

**#General comment on the comparison between sedimentary data and hydrological**

*Variables Q vs SSC are always presented as a log-log linear regressions. The same should applied to DLT vs Q, likely to P99D0 vs Q. From the scatterplot presented in Fig 8, it is likely that the general proxy-hydrometric relation follows a DLT=f(log(Q)), or a log(DLT)=f(log(Q) relation rather than a linear relation. See Warrick (2015) and references therein, or Thurston et al. (2020). This should be tested as it has major implications on statistical yields in the sediment-hydrological relations.*

*Warrick, J. A. (2015). Trend analyses with river sediment rating curves. Hydrological Processes, 29, 936–949. https://doi.org/10.1002/hyp.10198*
*Thurston et al. (2020). Modelling suspended sediment discharge in a glaciated Arctic catchment–Lake Peters, Northeast Brooks Range, Alaska.*
https://doi.org/10.1002/hyp.13846

*Reply:*
*Thank you for that comment and literatures. This will be tested.*

Authors' additional reply:

Variables DLT and $P99D_0$ vs Q were tested with a log-log linear regression. It improved very slightly the statistical results but had no major impacts (right scatterplot). Therefore, we are still proposing our reconstructions in cubic meters per second.

[Figure]

**#General comment on the regionalization of the signal**

*The merging of the different watersheds of the region is interesting, but I don't think that the quantitative analysis is relevant. This is exemplified by the low correlation of r=0.49 (even though significant) between the Naskaupi River and the Eagle station. This means that the discharge data from the Naskaupi River can only explain 24% of the variance in Eagle discharge data, independently from the sediment context. Removing Eagle from this merging exercise will not solve this issue. Each watershed is sensitive in its own way not only to specific climatic (evidence is missing that the climate in the Naskaupi region is representative of a broader region, not only through correlation between hydrometric station data) but also to geomorphic conditions that are not integrated into the daily climatic series of the CemaNeigeGR4J model (such as slope, erosion susceptibility, potential geological difference, orientation. . .), and that can differ significantly within the 500x500km grid used in this manuscript. A more detailed analysis of the different watershed, their runoff response (timing, strength, duration, sensitivity to snowmelt vs rainfall) would merit further investigation. L241: "These four streamflow series (Tab. 2) show strong positive correlations with Naskaupi River discharge", one expects to see these strong positive correlations. Figure 3 presenting the location of the different catchment for regionalization of the findings would have benefited an additional panel with daily streamflow time series for each catchment as in Figure 2, for instance.*

*Reply:*

*This is an excellent suggestion. Additional panels on Fig. 3 (streamflow regime for each catchment as in Figure 2 and series of annual streamflow anomalies from all hydrometric stations used in this study) will help discuss the similarity between different watersheds and justify the used of our regional instrumental series.*

[Figure]

*The daily climatic series used to build our Labrador region mean annual discharge series does not come from the rainfall-runoff model (CemaNeigeGR4J) but rather from instrumental data from hydrometric stations. We will make sure to improve the clarity of that section.*

Authors' modification:

Additional panels were added in Fig. 3. Discussion on the similarity between the hydrology of the different watersheds in Labrador was added in section 4.5, line 638.

**General comments on the calibration-in-time model**

*A proxy-hydrology calibration model is built for the period 1978-2011, and reconstructed back to 1876. Post 1972 (River deviation) shows that the system has changed hydrologically with discharge reduced by a factor 2. This should also be true sedimen- tologically, and a few points are in line with this (contre-)hypothesis: clear change in the preservation of DLT in NAS-2, change in the mean P99D0 record of NAS-1, change in mean and variance of DLT and TVT of BEA-1, and most significant change for TVT and*

*DLT post 1972 in NAS-2. These observations thus contradict the sentence L580 "River sediment input seems to have been quantitatively and spatially constant." The principle of stationarity being not respected hydrologically, it is doubtful that the calibration model post anthropogenic modification remains valid for the preceding period. Deeper discussion are required on this topic, e.g., by proposing evidence that the sediment record (through TVT, DLT, or best P99D0) is not significantly affected by this change and can be used to infer river hydrodynamics prior 1972.*

*Reply:*

*There is no instrumental data available for the Naskaupi basin before 1972. Thus, it is not possible to calibrate the model for the 1856-1971 period. We will further discuss the limitation and weakness of our calibration model in the revised version of the paper.*

*Our text in lines L580 explains that "River sediment input seems to have been quantitatively and spatially constant." Here we are talking about the 1856-1971 period (Fig. 6). This sentence does not apply for the period after the Naskaupi River diversion. This section will be clarified.*

*The diversion of the Naskaupi River caused certain changes in the sediment dynamics but did not modify it drastically. Despite the observed post-diversion changes in varve's parameters, the varves still respond directly to the river discharge. The part of the watershed that has been diverted is a section composed mainly of lakes which are not very hydrologically reactive.*

*The BEA core records inputs from the Beaver River, an adjacent watershed devoid of anthropogenic modifications. By integrating the BEA core into the pooled data, it helps to improve the natural hydrological signal in our mean series used for reconstruction.*

Authors' modification:

The limitation of our calibration model is now mentioned in section 5.2, line 973.

Despite the observed post-diversion changes in varves' physical parameters in cores NAS-1 and NAS-2, the varves still responded directly to variations in river discharge. The upper part of TVT and DLT series in core NAS-1 (1972-2016) show the most perceptible differences after 1972. This is the reason why this section was discarded from the combined DLT series used to reconstruct Q-mean to remove the likely anthropogenic impact on sedimentation during this period. There is also an increase of $P99D_0$ values in core NAS-1 after 1972, but this increase remains very moderate (see Supplement Fig. S3, and Tab. S3). We think that $P99D_0$ is not significantly affected by the Naskaupi River diversion and can be used to infer Q-max prior to 1972.

The sentence: "River sediment input seems to have been quantitatively and spatially constant." was removed from the text (line 913).

The integration of BEA-1 in the combined series is discussed in section 5.3, line 1009. In our opinion, this allows to capture the hydrological signal from a larger region (Nakaupi + Beaver watersheds).

*L604: "it is quite unlikely that the sedimentary input from the Naskaupi River contributed to sediment accumulation at the mouth of the Beaver River" is in contradiction with L440 : "data from core BEA-1 (1856-440 2016), NAS-1 (1856-2016) and NAS-2 (1968-2016) have been normalized and averaged to produce mean TVT, DLT and P99D0 series" to be compared to the Naskaupi River hydrometric station. This questions the selection of BEA-1 in the merging approach of the sedimentary data.*

*Reply:*
*Data from the Naskaupi River hydrometric station are considered to be also valid for the Beaver River due to the proximity of those watersheds. There are no instrumental data available for the Beaver River. Even if the core BEA does not directly record the inputs of the Naskaupi, it records the very similar inputs of the Beaver. This section will be clarified.*

Authors' modification:
As mentioned above, the integration of BEA-1 in the combined series is discussed in section 5.3, line 1009.

*Moreover, the justification that Naskaupi River discharge does not affect BEA-1 location is made by the fact that (L598-608) "the absence of any traces of the 1972 CE marker bed at the Beaver River mouth (BEA-1) supports this hypothesis." This argument is not admissible, especially with regards to the previous discussion (L583) that the "flood(s) of the years 1972 CE has (have) remobilized newly available sediments and deposited a thick and coarse-grained turbidite on the lake floor". It is indeed likely, with regards to the sedimentary facies of cores NAS, that the 1972 flood transported coarse material that plunged in the river proximal and extended as hyperpicnal flow following the lake Bathymetry (NAS-1 to NAS-2), thus not affecting BEA core location. However, discussion about flood hydrodynamics and annual river discharge in terms of sediment transport should be decoupled in the discussion.*

*Reply:*
*OK, this section will be clarified.*

Authors' modification:
This section was modified, line 918.

*The argument that a decline in varve thickness is also observed post 1972 in BEA, thus related to a natural hydro-climatic signal can be true, but seems superimposed to the effect of the Naskaupi River diversion, especially for cores NAS. While discreet peaks of sediment proxy (TVT, DLT, P99D0) for the different sediment cores are consistent (occurring at the same date), the variance, mean, and trend in these data are not comparable enough to allow the merging. Also, the three records from the three cores respond totally differently to the pos-1972 hydrological changes: lower mean for BEA-1, higher mean and increase variance for NAS-1, lower mean + decreased variance + decreasing trend for NAS-2. Suggestion: change point analysis (mean, variance and trend) can be performed on each times series, both from the hydrological and sedimentary variables. This would give statistical support to visual information.*

Reply:
We will add statistical supports to discuss the different response to post-1972 hydrological changes between cores.

Authors' modification:

The Supplement now provides additional visual support and statistical information on varve's parameters series. Quantitative data on the sedimentological response of cores NAS-1 and NAS-2 to post-1972 Naskaupi River hydrological changes are also available. These supplements show that total varve thickness (TVT), detrital layer thickness (DLT) and the particle size (P99D$_0$) series from different sites (BEA-1, NAS-1 and NAS-2) share similarities in their short- and longer-term variability, that help justified the combination of sedimentological data from different sites (combined series).

*Finally, I am really surprised to see a 5-year running mean for the reconstruction of hydrological data. As the varve chronology is more than robust, through its coherence between the different locations and perfect correspondence with 137Cs, it is a pity that annual time series are not reconstructed. This choice of smoothing the data needs to be justified. Running mean in lake sediment studies are generally used to account for the error in the varve chronology, with statistical justification for significant improvement of the proxy-climate correlations (cf. Von Gunten et al., 2012). + Figure 10 compares the rainfall-runoff model and sedimentary data at annual resolution, with no lag (L624). This gives again the impression that correlation values are maximized at all cost.*

*von Gunten, L., Grosjean, M., Kamenik, C. et al. Calibrating biogeochemical and physical climate proxies from non-varved lake sediments with meteorological data: methods and case studies. J Paleolimnol 47, 583–600 (2012).*
*https://doi.org/10.1007/s10933-012-9582-9*

*Reply:*

*The running mean was used to help the reader to visualize the low frequency hydrological variability, but it was not used to make the correlation. The annual time series (Q-mean and Q-max) are indeed reconstructed. We will consider removing the running mean from Fig. 8, 9.*

Authors' modification:

Considering the reply above, the 5- years running mean is still presented for the reconstructed annual Q-mean and Q-max time series in Fig. 8, 9.

The sentence: ''Cross correlations between varve parameter series (1856-2016) with instrumental data (1969-2011) and rainfall-runoff modeling reconstructions (1880-2011) show no lag, which demonstrates the accuracy of the time series used in this study'' (line 999) was removed from the text to focus on the main discussion.

**#General comment on the rainfall-runoff modeling approach**

*A key point of this review is the comparison between sedimentary data and modeling. The rainfall-runoff modeling for each catchment is merge to a single ANATEM time series (Fig. 10) and compared to the sediment properties of the varves. This ANATEM time series is based on the pre-determination of single catchment area, then extended for the whole studied period. However, the Naskaupi river watershed pre- and post-1972 is different (smaller after the 1972 river deviation) and should be adapted in the modeling; producing two time series (i) 1880-1971, (ii) 1973-2011. This likely explains that stronger correlations found between e.g., DLT and ANATEM for the period 1972-2011 (r=0.54) compared to the preceding period (r=0.31).*

*Reply:*

*There is some sort of misunderstanding here. The rainfall-runoff modeling was not performed on each catchment and merge to a single ANATEM time series. The rainfall-runoff modeling was solely performed with the Naskaupi River hydrometric station area (Fig. 10). This will be clarified in the revised version.*

*As mentioned above, there is no instrumental data available for the Naskaupi basin before 1972. So, it is not possible to calibrate the modelling for the 1856-1971 period…*

Authors' modification:

It is now specified that the rainfall-runoff modelling was performed for the Naskaupi River hydrometric station area (section 3.6).

**#Specific comments**

*L68: to reconstruct daily. . .*

> *Reply:*
>
> *OK*

> Authors' modification:
>
> Done, line 72.

*L73: "Long hydro-climatic series based on natural proxies in the study region are rare and limited to tree-ring". What have all these studies produced? What conclusions ? Is the aim of the present study to comfort previous finding, to increase spatial coverage? This does not say why clastic lake sediment are better than tree rings or pollen data (which is suggested here) Aren't tree-ring records not enough? Are they all from the Labrador region? Are the hydroclimate records consistent with each others? Answering these question would help re-shaping the sentences in explaining what makes clastic varves so specific and powerful.*

> *Reply:*
>
> *This is an excellent suggestion. This will be done in the revised version.*

> Authors' modification:
>
> The introduction has been improved according to this suggestion, line 79 to 93.

*L76: "clastic" are not defined prior to this mention*

> *Reply:*
>
> *Clastic will be defined.*

> Authors' modification:
>
> "Clastic" was defined (line 89).

*L79: Remover 'The' between area and into*

> *Reply:*
>
> *OK*

> Authors' modification:
>
> Done, line 95.

*L81: Amann et la., should be et al.,*

> *Reply:*
>
> *OK*

> Authors' modification:
>
> Done, line 97.

*L231 : remove 'used' form the title*

> *Reply:*
>
> *OK*

> Authors' modification:
>
> Done, line 305.

*L245: Suggested change; 'This allows to extend instrumental data series for the period 1969 to 2011, and fill in data for the missing years.'*

> *Reply:*
>
> *OK*

> Authors' modification:
>
> Done, line 317.

*L252: title could be simply, e.g., varve properties and hydrological variables*

> *Reply:*
>
> *OK*

> Authors' modification:
>
> This title was simplified (line 334).

*L456: "data show significant (p < 0.01) strong positive correlation." Remove 'strong', especially referring to r = 0.49 in brackets, this is not a strong correlation, especially in such hydrological context.*

> *Reply:*
>
> *OK*

> Authors' modification:
>
> Done, line 643.

*L478: " The significant correlation between reconstructed Q-mean and Q-max values and observed discharge data validates the predictive capacity of the model." I don't see how the fact that Qmean and Qmax correlates validates the proxy-Q model.*

*Reply:*

*This sentence will be changed.*

Authors' modification:

This sentence was removed when modifying section 4.6.

*L496: "demonstrates that the Grand Lake varved sequence is robust and contains a regional signal." You mean the hydrological reconstruction is robust? I would not say that R2 = 0.41 is robust. Remove 'robust' and keep 'contains a regional signal. Q-mean and Q-max are sometimes written with a capital M (e.g., Q-Mean), sometimes not (Q-mean). Please stay consistent.*

*Reply:*

*This will be done.*

Authors' modification:

Done, line 746.

*L595: it should read 'indicate that the capacity of spring discharge to transport fine sediment and its ability to float ice to Grand Lake decreases due to the decrease in water supply.'*

*Reply:*

*OK*

Authors' modification:

Done, line 955.

*L650: please consider changing 'robust' for 'best proxy'*

*Reply:*

*This will be done.*

Authors' modification:

Done, line 1059.

*L697: extracted from*

> *Reply:*
> *OK*

> Authors' modification:
> Done, line 1139.

*L706: change "could help to better our reconstructions" to 'could help better refine these reconstructions"*

> *Reply:*
> *OK*

> Authors' modification:
> Done, line 1175.

[revised manuscript text omitted]

<table>
<tr><td>**Supprimé:** , for each studied catchment,</td></tr>
<tr><td>**Supprimé:** ,</td></tr>
<tr><td>**Supprimé:** for each catchment,</td></tr>
<tr><td>**Supprimé:** sub-</td></tr>
<tr><td>**Supprimé:** sub-</td></tr>
<tr><td>**Supprimé:** varve</td></tr>
<tr><td>**Supprimé:** The</td></tr>
<tr><td>**Supprimé:** of the detrital layer</td></tr>
<tr><td>**Supprimé:** D</td></tr>
<tr><td>**Supprimé:** occasionally</td></tr>
<tr><td>**Supprimé:** s</td></tr>
<tr><td>**Supprimé:** in some cases</td></tr>
<tr><td>**Supprimé:** non-annual</td></tr>
<tr><td>**Supprimé:** The</td></tr>
<tr><td>**Supprimé:** abundance</td></tr>
<tr><td>**Supprimé:** varve</td></tr>
<tr><td>**Supprimé:** sub-</td></tr>
</table>

[revised manuscript text omitted]

---

## Author Response (AR2)

**cp-2020-87**

**Reconstructing past hydrology of eastern Canadian boreal catchments using clastic varved sediments and hydro-climatic modelling: 160 years of fluvial inflows**

By Antoine Gagnon-Poiré, Pierre Brigode, Pierre Francus, David Fortin, Patrick Lajeunesse, Hugues Dorion and Annie-Pier Trottier.

**Point-by-point reply to the referee #2 comments**

The review made by the referee #2 is gratefully acknowledged. The manuscript has been modified in response to the comments. In the following document, the referee comments are listed in italic blue, the specific responses to referee comments given by the authors are in black. The line numbers mentioned in section "reply", refer to the line numbers in the Marked-up manuscript. Suggested minor changes referring to a comment from the previous report are proposed at the end of this document.

**REFEREE #2**

**General comments:**

*The manuscript by Gagnon-Poiré and co-authors entitled 'Reconstructing past hydrology of eastern Canadian boreal catchments using clastic varved sediments and hydro-climatic modeling: 160 years of fluvial inflows' has been revised according to reviewer' comments and a detailed point-by-point response to reviewer comments is presented. Most comments are adequately considered by the authors, but there are still two points that could be better clarified.*

> *Reply:*
> *We thank reviewer #2 for his positive comments on our revised manuscript.*

**Specific comments:**

*First, I agree to the argument that by pooling more than one sediment record local disturbances of the sediment record might be minimized and I appreciate especially the approach to combine a distal and proximal core for the part of the lake influenced by the Naskaupi (NAS-1 and 2). However, it would have been logical to apply the same approach with a distal and a proximal core also for the Beaver River in order to get both parts of the large catchment similarly represented by sediment records. In this respect, it should be clarified why only one core representing the Beaver is included in the analyses and how this different representation of catchments in sediment records influences the analyses.*

> Reply:
>
> *The main tributary of Grand Lake is the Naskaupi River. Considering that the watershed of the Naskaupi river is twice as large as that of the Beaver River, we estimated that the Beaver River's discharge is at least two times less than the Naskaupi River discharge. The Beaver River delta slopes and associated sediment waves are much less developed than at the mouth of the Naskaupi River, which also shows that the Beaver River is the secondary tributary of Grand Lake. So, we believe that the site BEA-1 alone is sufficient to represent the signal for this Grand Lake's sub-watershed. Our goal during the coring mission carried out in 2017 was to focus on the hydrological signal of the Naskaupi River watershed as the hydrometric station 03PB002 measuring the discharge is present in this specific Grand Lake sub-catchment. The purpose of collecting a core at the mouth of the Beaver River was to sample this smaller adjacent Grand Lake's sub-catchment, which is devoid of anthropogenic modifications.*
>
> *As mentioned in section 3.1, site BEA-1 and NAS-1 were collected from locations sharing relative similarities (at the distal frontal slope of the Beaver and Naskaupi river deltas; fig. 1c) while site NAS-2 was collected away from the Naskaupi River delta, at the beginning of the deep lake basin. We think that the sites NAS-1 and NAS-2 record the Naskaupi River hydrologic signal, but are also susceptible to record the Beaver River signal too, due to their location in the axis of both rivers. Yet, NAS-2 and NAS-1 can be considered as distal cores from the beaver River as well. It is also to consider that we used the DLT and the P99 for our final reconstructions. Those proxies have been extracted from thin sections for the last 160 years (1856-2016) for cores BEA-1 and NAS-1, and only for the last 47 years (1968-2016) for core NAS-2. Thus, the main coring sites, which are BEA-1 and NAS-1, cover the entire length of the combined series while the NAS-2 site contributed only to the surface of the combined series.*
>
> Text have been added (lines 177 to 182) to provide additional information on the location of the coring sites.

*Second, the effect of the Naskaupi River diversion on sedimentation in the NAS cores as described in lines 670-687 in the manuscript still is not entirely clear. I understood that sedimentation increased due to more sediments available in the catchment due to the changes caused by constructions. It is also stated that the NAS-1 site became 'more sensitive to maximum discharge variations in spring than mean annual discharges' (lines 675-676). This would make sense, but apparently this applies only for parts of the spring runoff because 'the capacity of early spring discharge to transport fine sediments… decreases along with the decrease in water supplies'. This apparent contradiction and differentiation in 'early spring' and 'spring' should be better explained. Furthermore, the different behavior of site NAS-2, where sediment input declined after 1971 in contrast to NAS-1 where it increased, could be better explained. Is that due to the lower transport capacity of the discharge and does it mean that the additional sediment is mainly accumulated on the delta, thus changing the entire delta geometry?*

*Reply:*

We agree with those comments. We modified the text to better highlight the difference between the fine early spring layers associated to the 'early spring discharge' and the detrital layers mainly related to the 'maximum spring discharge' (lines 694 to 711). We also proposed the potential mechanism for the recent (post-1971) difference in sedimentation between the site NAS-1 and NAS-2 (lines 719 to 725).

**We would also like to make some additional minor modifications to better address a comment made reviewer #2 in her/his former report.**

''*In general, it is difficult to follow the large number of different statistical correlations between cores, proxies, proxy reconstruction and model results. A more concise approach with a focus on main correlations would make the manuscript easier to read.*'',

While working on this revision, we realized that correlations with climatic variables were seldom described and discussed and were less relevant. Therefore, we propose to focus our study on the relations between hydrological variables (calculated from the time series of daily discharge) with Grand Lake's varved sequences. We suggest to drop the presentations and the discussion about the correlations between the few climatic variables (winter snowfall, spring temperature, and rainfall) with varve parameters. Deletions proposed here are far from being substantial and are based on the removal of only 80 words from the manuscript. The suggested cuts related this suggestion are highlighted in yellow in the marked-up manuscript.

We consider that removing this information does not affect the quality and the interpretation of our dataset, as well as the scientific content of the manuscript. On the contrary, it allows to better focus on the hydrological series contained in the varved record, which makes the manuscript simpler and easier to read as initially suggested by the comments of referee #2. We are aware that these changes could have been proposed previously, and we are open to disregard these modifications if the editor's is not comfortable with them.